# Efficient Training of Energy-Based Models Using Jarzynski Equality

**Davide Carbone**
Dipartimento di Scienze Matematiche, Politecnico di Torino
Istituto Nazionale di Fisica Nucleare, Sezione di Torino
davide.carbone@polito.it

**Mengjian Hua**
Courant Institute of Mathematical Sciences, New York University
mh5113@nyu.edu

**Simon Coste**
LPSM, Université Paris-Cité
simon.coste@u-paris.fr

**Eric Vanden-Eijnden**
Courant Institute of Mathematical Sciences, New York University
eve2@nyu.edu

## Abstract

Energy-based models (EBMs) are generative models inspired by statistical physics with a wide range of applications in unsupervised learning. Their performance is well measured by the cross-entropy (CE) of the model distribution relative to the data distribution. Using the CE as the objective for training is however challenging because the computation of its gradient with respect to the model parameters requires sampling the model distribution. Here we show how results for nonequilibrium thermodynamics based on Jarzynski equality together with tools from sequential Monte-Carlo sampling can be used to perform this computation efficiently and avoid the uncontrolled approximations made using the standard contrastive divergence algorithm. Specifically, we introduce a modification of the unadjusted Langevin algorithm (ULA) in which each walker acquires a weight that enables the estimation of the gradient of the cross-entropy at any step during GD, thereby bypassing sampling biases induced by slow mixing of ULA. We illustrate these results with numerical experiments on Gaussian mixture distributions as well as the MNIST and CIFAR-10 datasets. We show that the proposed approach outperforms methods based on the contrastive divergence algorithm in all the considered situations.

## 1 Introduction

Probabilistic models have become a key tool in generative artificial intelligence (AI) and unsupervised learning. Their goal is twofold: explain the training data, and allow the synthesis of new samples. Many flavors have been introduced in the last decades, including variational auto-encoders [1, 2, 3] generative adversarial networks [4, 5], normalizing flows [6, 7, 8, 9, 10], diffusion-based models [11, 12, 13], restricted Boltzmann machines [14, 15, 16], and energy-based models (EBMs) [17, 18, 19].

37th Conference on Neural Information Processing Systems (NeurIPS 2023).

Inspired by physics, EBMs are unnormalized probability models, specified via an energy function $U$, with the underlying probability density function (PDF) defined as $\rho = \exp(-U)/Z$, where $Z$ is a normalization constant: this constant is often intractable but crucially it is not needed for data generation via Monte-Carlo sampling. In statistical physics, such PDFs are called Boltzmann-Gibbs densities [20, 21, 22] and their energy function is often known in advance; in the context of EBMs, the aim is to estimate this energy function in some parametric class using the available data. Such models benefit from a century of intuitions from computational physics to guide the design of the energy $U$, and help the sampling; hence, interpretability is a clear strength of EBMs, since in many applications the parameters of the model have a direct meaning. Unfortunately, training EBMS is a challenging task as it typically requires to sample complex multimodal (i.e. non-log-concave) distributions in high-dimension.

This issue arises because real-world datasets are often clustered into different modes with imbalanced weights. A skewed estimation of these weights can lead to harmful biases when it comes to applications: for example, under-representation of elements from a particular population. It is important to ensure that these biases are either avoided or compensated, leading to methods that ensure fairness ([23]); however, the training routines for EBMs often have a hard time properly learning the weights of different modes, and can often completely fail at this task.

Two popular classes of methods are used to train an EBM. In the first, the Fisher divergence between the model and the data distributions is taken as the objective to minimize, which amounts to learning the gradient of the energy function, $\nabla U = -\nabla \log \rho$. Although computationally efficient due to score-matching techniques [24], methods in this class are provably unable to learn the relative weights of modes when they are separated by low-density regions.

In the second class of methods, the cross-entropy between the model and the data distributions is used as the objective. Unlike the Fisher divergence, the cross-entropy is sensitive to the relative weights in multimodal distributions, but it unfortunately leads to training algorithms that are less justified theoretically and more delicate to use in practice. Indeed, gradient-based optimization procedures on the cross-entropy require sampling from the model distribution using Markov-Chain Monte-Carlo (MCMC) methods or the unadjusted Langevin algorithm (ULA) until mixing, which in a high-dimensional context or without log-concavity can be prohibitive, even considering the large computational power available today.

As a result, one typically needs to resort to approximations to estimate the gradient of the cross-entropy, for example by using the contrastive divergence (CD) or the persistent contrastive divergence (PCD) algorithms, see [25, 26, 27]. Unfortunately, the approximations made in these algorithms are uncontrolled and they are known to induce biases similar to those observed with score-based methods (the CD algorithm reduces to gradient descent over the Fisher divergence in the limit of frequent resetting from the data [28]). Many techniques have been proposed to overcome this issue, for example, based on MCMC sampling [29, 30, 31] or on various regularizations of CD [32, 33]. In practice though, these techniques still do not handle well multimodal distributions and they come with little theoretical guarantees.

**Main contributions.** In this work, we go back to the original problem of training EBMS using the cross-entropy between the model and the data distributions as objective, and:

- We derive exact expressions for the cross-entropy and its gradient that involve expectations over an ensemble of weighted walkers whose weights and positions evolve concurrently with the model energy during optimization. Our main tool is Jarzynski equality [34], an exact relation between the normalizing constants of two distributions linked by an out-of-equilibrium dynamics.

- Based on these formulas, we design sequential Monte Carlo sampling procedures [35] to estimate the cross-entropy and its gradient in a practical way at essentially no additional computational cost compared to using the CD or PCD algorithms with ULA.

- We show that reweighting the walkers is necessary in general and that training procedures based on the CD algorithm lead to uncontrolled biases whereas those based on using the PCD algorithm lead to mode collapse in general.

- We illustrate these results numerically on synthetic examples involving Gaussian mixture models where these effects can be demonstrated.

- We also apply our method on the MNIST and CIFAR-10 datasets. In the first, we intentionally bias the proportion of the different digits, to demonstrate that our method allows the retrieval of these proportions whereas the other methods fail to do so.

**Related works.** How to train EBMs is a longstanding question, and we refer e.g. to [17, 36] for overviews on this topic. The use of modern deep neural architectures to model the energy was then proposed in [37], and a relation between EBMs and classifiers has been highlighted in [38]. How to sample from unnormalized probability models is also an old and rich problem, see [39, 40] for general introductions.

Score-matching techniques and variants originate from [24, 41, 42]; their shortcoming in the context of EBM training is investigated in [43] and their blindness to the presence of multiple, imbalanced modes in the target density has been known for long: we refer to [44, 45] for discussions. Contrastive divergence (CD) algorithms originate from [25, 26, 27]. These methods only perform one or a few sampling steps of the algorithm, with random walkers that are repeatedly restarted at the data points. Persistent contrastive divergence (PCD) algorithm, introduced in [29], eliminates the restarts and evolves walkers using ULA. Unlike the approach proposed in this paper, these methods are known to give estimates of the gradient of the cross-entropy that have uncontrolled biases which are difficult to remove; there were many attempts in this direction, also in cooperation with other unsupervised techniques, see e.g. [46, 47, 48, 49, 50, 51, 52].

It is worth noting that the original papers on CD proposed to use an objective which is different from the cross-entropy, and their proposed implementation (what we call the CD algorithm) does not perform gradient descent on this objective, due to some gradient terms being neglected [33]. This sparked some debate about which objective, if any, was actually minimized by CD algorithms; for example, [53, 54] showed that the CD and PCD algorithms are essentially adversarial procedures; [33] introduced a way to approximate the missing term in the gradient of the CD objective; and [28, 55] showed that in the limit of small noise, CD is essentially equivalent to score matching. In contrast, there is no ambiguity about the objective used in the method we propose: it is always the cross-entropy.

Jarzynski equality (JE) was introduced in [34] and gives an exact expression relating the normalizing constants (or equivalently free energy [22]) of two distributions linked by out-of-equilibrium continuous-time dynamics. A discrete-time analog of JE is used in Neal's annealed importance sampling [56], which belongs to the framework of sequential Monte-Carlo methods [35]. These methods have been used in the context of generative models based on variational autoencoders [57, 58], normalizing flows [59, 60], and diffusion-based models [61, 62]. In contrast, here we use sequential Monte-Carlo method to train EBM on the cross-entropy directly.

## 2 Energy-Based Models

**Setup, notations, and assumptions.** The problem we consider can be formulated as follows: we assume that we are given $n \in \mathbb{N}$ data points $\{x_i^*\}_{i=1}^n$ in $\mathbb{R}^d$ drawn from an unknown probability distribution that is absolutely continuous with respect to the Lebesgue measure on $\mathbb{R}^d$, with a positive probability density function (PDF) $\rho_*(x) > 0$ (also unknown). Our aim is to estimate this PDF via an energy-based model (EBM), i.e. to find a suitable energy function in a parametric class, $U_\theta : \mathbb{R}^d \to [0, \infty)$ with parameters $\theta \in \Theta$, such that the associated Boltzmann-Gibbs PDF

$$\rho_\theta(x) = Z_\theta^{-1} e^{-U_\theta(x)}; \qquad Z_\theta = \int_{\mathbb{R}^d} e^{-U_\theta(x)} dx \qquad (1)$$

is an approximation of the target density $\rho_*(x)$. The normalization factor $Z_\theta$ is known as the partition function in statistical physics [22]. This factor is hard to estimate and one advantage of EBMs is that they provide generative models that do not require the explicit knowledge of $Z_\theta$ since Markov Chain Monte-Carlo (MCMC) methods can in principle be used to sample $\rho_\theta$ knowing only $U_\theta$ – the design of such MCMC methods is an integral part of the problem of building an EBM.

To proceed we will assume that the parametric class of energy we use is such that, for all $\theta \in \Theta$,

$$U_\theta \in C^2(\mathbb{R}^d); \qquad \exists L \in \mathbb{R}_+ : \|\nabla\nabla U_\theta(x)\| \leq L \quad \forall x \in \mathbb{R}^d;$$
$$\exists a \in \mathbb{R}_+ \text{ and a compact set } \mathcal{C} \in \mathbb{R}^d : \ x \cdot \nabla U_\theta(x) \geq a|x|^2 \quad \forall x \in \mathbb{R}^d \setminus \mathcal{C}. \qquad (2)$$

These assumptions guarantee that $Z_\theta < \infty$ (i.e. we can associate a PDF $\rho_\theta$ to $U_\theta$ via (1) for any $\theta \in \Theta$) and that the Langevin equations as well as their time-discretized versions we will use to sample $\rho_\theta$ have global solutions and are ergodic [63, 64, 65]. We stress that (2) *does not* imply that $U_\theta$ is convex (i.e. that $\rho_\theta$ is log-concave): in fact, we will be most interested in situations where $U_\theta$ has multiple local minima so that $\rho_\theta$ is multimodal. For simplicity we will also assume that $\rho_*$ is in the parametric class, i.e. $\exists \theta_* \in \Theta : \rho_{\theta_*} = \rho_*$. Our aims are primarily to identify this $\theta_*$ and to sample $\rho_{\theta_*}$; in the process, we will also show how to estimate $Z_{\theta_*}$.

**Cross-entropy minimization.** To measure the quality of the EBM and train its parameters one can use the cross-entropy of the model density $\rho_\theta$ relative to the target density $\rho_*$

$$H(\rho_\theta, \rho_*) = -\int_{\mathbb{R}^d} \log \rho_\theta(x) \rho_*(x) dx = \log Z_\theta + \int_{\mathbb{R}^d} U_\theta(x) \rho_*(x) dx \tag{3}$$

where we used the definition of $\rho_\theta$ in (1) to get the second equality. The cross-entropy is related to the Kullback-Leibler divergence via $H(\rho_\theta, \rho_*) = H(\rho_*) + D_{\text{KL}}(\rho_* || \rho_\theta)$, where $H(\rho_*)$ is the entropy of $\rho_*$, and its gradient with respect to the parameter $\theta$ can be calculated using the identity $\partial_\theta \log Z_\theta = -\int_{\mathbb{R}^d} \partial_\theta U_\theta(x) \rho_\theta(x) dx$, to obtain

$$\begin{aligned} \partial_\theta H(\rho_\theta, \rho_*) &= \int_{\mathbb{R}^d} \partial_\theta U_\theta(x) \rho_*(x) dx - \int_{\mathbb{R}^d} \partial_\theta U_\theta(x) \rho_\theta(x) dx \\ &\equiv \mathbb{E}_*[\partial_\theta U_\theta] - \mathbb{E}_\theta[\partial_\theta U_\theta]. \end{aligned} \tag{4}$$

The cross-entropy is more stringent, and therefore better, than other objectives like the Fisher divergence: for example, unlike the latter, it is sensitive to the relative probability weights of modes on $\rho_*$ separated by low-density regions [36]. Unfortunately, the cross entropy is also much harder to use in practice since evaluating it requires estimating $Z_\theta$, and evaluating its gradient requires calculating the expectation $\mathbb{E}_\theta[\partial_\theta U_\theta]$ (in contrast $\mathbb{E}_*[U_\theta]$ and $\mathbb{E}_*[\partial_\theta U_\theta]$ can be readily estimated on the data). Typical training methods, e.g. based on the CD or the PCD algorithms, give up on estimating $Z_\theta$ and resort to various approximations to calculate the expectation $\mathbb{E}_\theta[\partial_\theta U_\theta]$—see Appendix A.5 for more discussion about these methods. While these approaches have proven successful in many situations, they are prone to training instabilities that limit their applicability. They also come with no theoretical guarantees in terms of convergence.

## 3 Training via sequential Monte-Carlo methods based on Jarzynski equality

In this section we use tools from nonequilibrium statistical mechanics [34, 56] to write exact expressions for both $\mathbb{E}_\theta[\partial_\theta U_\theta]$ and $Z_\theta$ (Sec 3.1) that are amenable to empirical estimation via sequential Monte-Carlo methods [35], thereby enabling gradient descent-type algorithms for the optimization of EBMs (Sec. 3.2).

### 3.1 Jarzynski equality in discrete-time

**Proposition 1.** *Assume that the parameters $\theta$ are evolved by some time-discrete protocol $\{\theta_k\}_{k \in \mathbb{N}_0}$ and that (2) hold. Given any $h \in (0, L)$, let $X_k \in \mathbb{R}^d$ and $A_k \in \mathbb{R}$ be given by the iteration rule*

$$\begin{cases} X_{k+1} = X_k - h\nabla U_{\theta_k}(X_k) + \sqrt{2h}\xi_k, & X_0 \sim \rho_{\theta_0}, \\ A_{k+1} = A_k - \alpha_{k+1}(X_{k+1}, X_k) + \alpha_k(X_k, X_{k+1}), & A_0 = 0, \end{cases} \tag{5}$$

*where $U_\theta(x)$ is the model energy, $\{\xi_k\}_{k \in \mathbb{N}_0}$ are independent $N(0_d, I_d)$, and we defined*

$$\alpha_k(x, y) = U_{\theta_k}(x) + \tfrac{1}{2}(y - x) \cdot \nabla U_{\theta_k}(x) + \tfrac{1}{4}h|\nabla U_{\theta_k}(x)|^2 \tag{6}$$

*Then, for all $k \in \mathbb{N}_0$,*

$$\mathbb{E}_{\theta_k}[\partial_\theta U_{\theta_k}] = \frac{\mathbb{E}[\partial_\theta U_{\theta_k}(X_k) e^{A_k}]}{\mathbb{E}[e^{A_k}]}, \qquad Z_{\theta_k} = Z_{\theta_0} \mathbb{E}[e^{A_k}] \tag{7}$$

*where the expectations on the right-hand side are over the law of the joint process $(X_k, A_k)$.*

---

**Algorithm 1** Sequential Monte-Carlo training with Jarzynski correction

---

1: **Inputs:** data points $\{x_*^i\}_{i=1}^n$; energy model $U_\theta$; optimizer step $\mathrm{opt}(\theta, \mathcal{D})$ using $\theta$ and the empirical CE gradient $\mathcal{D}$; initial parameters $\theta_0$; number of walkers $N \in \mathbb{N}_0$; set of walkers $\{X_0^i\}_{i=1}^N$ sampled from $\rho_{\theta_0}$; total duration $K \in \mathbb{N}$; ULA time step $h$; set of positive constants $\{c_k\}_{k\in\mathbb{N}}$.

2: $A_0^i = 0$ for $i = 1, \ldots, N$.

3: **for** $k = 0, \ldots, K-1$ **do**

4:      $p_k^i = \exp(A_k^i) / \sum_{j=1}^N \exp(A_k^j)$                       ▷ normalized weights

5:      $\tilde{\mathcal{D}}_k = \sum_{i=1}^N p_k^i \partial_\theta U_{\theta_k}(X_k^i) - n^{-1} \sum_{j=1}^n \partial_\theta U_{\theta_k}(x_*^j)$      ▷ empirical CE gradient

6:      $\theta_{k+1} = \mathrm{opt}(\theta_k, \tilde{\mathcal{D}}_k)$                                 ▷ optimization step

7:      **for** $i = 1, \ldots, N$ **do**

8:          $X_{k+1}^i = X_k^i - h\nabla U_{\theta_k}(X_k^i) + \sqrt{2h}\,\xi_k^i, \qquad \xi_k^i \sim \mathcal{N}(0_d, I_d)$      ▷ ULA

9:          $A_{k+1}^i = A_k^i - \alpha_{k+1}(X_{k+1}^i, X_k^i) + \alpha_k(X_k^i, X_{k+1}^i)$      ▷ weight update

10:      **end for**

11:      Resample the walkers and reset the weights if $\mathrm{ESS}_{k+1} < c_{k+1}$, see (13). ▷ resampling step

12: **end for**

13: **Outputs:** Optimized energy $U_{\theta_K}$; set of weighted walkers $\{X_K^i, A_K^i\}_{i=1}^N$ sampling $\rho_{\theta_K}$; partition function estimate $\tilde{Z}_{\theta_K} = Z_{\theta_0} N^{-1} \sum_{i=1}^N \exp(A_K^i)$; CE estimate $\log \tilde{Z}_{\theta_K} + n^{-1} \sum_{j=1}^n U_{\theta_K}(x_*^j)$

---

The proof of the proposition is given in Appendix A.2: for completeness we also give a continuous-time version of this proposition in Appendix A.1. We stress that the inclusion of the weights in (7) is key, as $\mathbb{E}[\partial_\theta U_{\theta_k}(X_k)] \neq \mathbb{E}_{\theta_k}[\partial_\theta U_{\theta_k}]$ in general. We also stress that (7) holds *exactly* despite the fact that for $h > 0$ the iteration step for $X_k$ in (5) is that of the unadjusted Langevin algorithm (ULA) with no Metropolis correction as in MALA [66, 67]. That is, the inclusion of the weights $A_k$ exactly corrects for the biases induced by both the slow mixing and the time-discretization errors in ULA.

Proposition 1 shows that we can evolve the parameters by gradient descent over the cross-entropy by solving (5) concurrently with

$$\theta_{k+1} = \theta_k + \gamma_k \mathcal{D}_k, \qquad \mathcal{D}_k = -\partial_\theta H(\rho_{\theta_k}, \rho_*) = \frac{\mathbb{E}[\partial_\theta U_{\theta_k}(X_k) e^{A_k}]}{\mathbb{E}[e^{A_k}]} - \mathbb{E}_*[\partial_\theta U_{\theta_k}], \quad (8)$$

where $\gamma_k > 0$ is the learning rate and $k \in \mathbb{N}_0$ with $\theta_0$ given. We can also replace the gradient step for $\theta_k$ in (8) by any update optimization step (via AdaGrad, ADAM, etc.) that uses as input the gradient $\mathcal{D}_k$ of the cross-entropy evaluated at $\theta_k$ to get $\theta_{k+1}$. Assuming that we know $Z_{\theta_0}$ we can track the evolution of the cross-entropy via

$$H(\rho_{\theta_k}, \rho_*) = \log \mathbb{E}[e^{A_k}] + \log Z_{\theta_0} + \mathbb{E}_*[U_{\theta_k}]. \quad (9)$$

### 3.2 Practical implementation

**Empirical estimators and optimization step.** We introduce $N$ independent pairs of walkers and weights, $\{X_k^i, A_k^i\}_{i=1}^N$, which we evolve independently using (5) for each pair. To evolve $\theta_k$ from some prescribed $\theta_0$ we can then use the empirical version of (8):

$$\theta_{k+1} = \theta_k + \gamma_k \tilde{\mathcal{D}}_k, \quad (10)$$

where $\tilde{\mathcal{D}}_k$ is the estimator for the gradient in $\theta$ of the cross-entropy:

$$\tilde{\mathcal{D}}_k = \frac{\sum_{i=1}^N \partial_\theta U_{\theta_k}(X_k^i) \exp(A_k^i)}{\sum_{i=1}^N \exp(A_k^i)} - \frac{1}{n} \sum_{j=1}^n \partial_\theta U_{\theta_k}(x_*^j), \quad (11)$$

These steps are summarized in Algorithm 1, which is a specific instance of a sequential Monte-Carlo algorithm. We can also use mini-batches of $\{X_k^i, A_k^i\}_{i=1}^N$ and the data set $\{x_*^j\}_{j=1}^n$ at every iteration (see Algorithm 2 in Appendix A.3), and switch to any optimizer step that uses $\theta_k$ and $\mathcal{D}_k$ as input to get the updated $\theta_{k+1}$. During the calculation, we can monitor the evolution of the partition function and the cross-entropy using as estimators

$$\tilde{Z}_{\theta_k} = Z_{\theta_0} \frac{1}{N} \sum_{i=1}^N \exp(A_k^i), \qquad \tilde{H}_k = \log \tilde{Z}_{\theta_k} + \frac{1}{n} \sum_{j=1}^n U_{\theta_K}(x_*^j) \quad (12)$$

These steps are summarized in Algorithm 1, which is a specific instance of a sequential Monte-Carlo algorithm. We adapted our routine to mini-batches in Algorithm 2 without any explicit additional source of error: in fact, the particles outside the mini-batch have their weights updated too. The only sources of error in these algorithms come from the finite sample sizes, $N < \infty$ and $n < \infty$. Regarding $n$, we may need to add a regularization term in the loss to avoid overfitting: this is standard. Regarding $N$, we need to make sure that the effective sample size of the walkers remains sufficient during the evolution. This is nontrivial since the $A_k^i$'s will spread away from zero during the optimization, implying that the weights $\exp(A_k^i)$ will become non-uniform, thereby reducing the effective sample size. This is a known issue with sequential Monte-Carlo algorithms that can be alleviated by resampling as discussed next.

**Resampling step.** A standard quantity to monitor the effective sample size [40] is the ratio between the square of the empirical mean of the weights and their empirical variance, i.e.

$$\text{ESS}_k = \frac{\left(N^{-1} \sum_{i=1}^N \exp(A_k^i)\right)^2}{N^{-1} \sum_{i=1}^N \exp(2A_k^i)} \in (0, 1] \tag{13}$$

The effective sample size of the $N$ walkers is $\text{ESS}_k N$. Initially, since $A_0^i = 0$, $\text{ESS}_0 = 1$, but it decreases with $k$. At each iteration $k_r$ such that $\text{ESS}_{k_r} < c_{k_r}$, where $\{c_k\}_{k \in \mathbb{N}}$ is a set of predefined positive constants in $(0, 1)$, we then:

1. Resample the walkers $X_{k_r}^i$ using $p_{k_r}^i = e^{A_{k_r}^i} / \sum_{j=1}^N e^{A_{k_r}^j}$ as probability to pick walker $i$;

2. Reset $A_{k_r}^i = 0$;

3. Use the update $Z_{\theta_k} = Z_{\theta_{k_r}} N^{-1} \sum_{i=1}^N \exp(A_k^i)$ for $k \geq k_r$ until the next resampling step.

This resampling is standard [35] and can be done with various levels of sophistication, as discussed in Appendix A.4. Other criteria, based e.g. on the entropy of the weights, are also possible, see e.g. [68].

**Generative modeling.** During the training stage, i.e. as the weights $A_k^i$ evolve, the algorithm produces weighted samples $X_k^i$. At any iteration, however, equal-weight samples can be generated by resampling. Notice that, even if we no longer evolve the model parameters $\theta_k$, the algorithm is such that it removes the bias from ULA coming from $h > 0$ – this bias removal is not perfect, again because $N$ is finite, but this can be controlled by increasing $N$ at the stage when the EBM is used as a generative model.

## 4 Numerical experiments

### 4.1 Gaussian Mixtures

In this section, we use a synthetic model to illustrate the advantages of our approach. Specifically, we assume that the data is drawn from the Gaussian mixture density with two modes given by

$$\rho_*(x) = Z_*^{-1} \left(e^{-\frac{1}{2}|x-a_*|^2} + e^{-\frac{1}{2}|x-b_*|^2 - z_*}\right), \qquad Z_* = (2\pi)^{d/2} \left(1 + e^{-z_*}\right) \tag{14}$$

where $a_*, b_* \in \mathbb{R}^d$ specify the means of the two modes and $z_* \in \mathbb{R}$ controls their relative weights $p_* = 1/(1 + e^{-z^*})$ and $q_* = 1 - p_* = e^{-z_*}/(1 + e^{-z^*})$. The values of $a_*, b_*, z_*$ are carefully chosen such that the modes are well separated and the energy barrier between the modes is high enough such that jumps of the walkers between the modes are not observed during the simulation with ULA. Consistent with (14) we use an EBM with

$$U_\theta(x) = -\log\left(e^{-\frac{1}{2}|x-a|^2} + e^{-\frac{1}{2}|x-b|^2 - z}\right), \tag{15}$$

where $\theta = (a, b, z)$ are the parameters to be optimized. We choose this model as it allows us to calculate the partition function of the model at any value of the parameters, $Z_\theta = (2\pi)^{d/2} \left(1 + e^{-z}\right)$. We use this information as a benchmark to compare the prediction with those produced by our method.

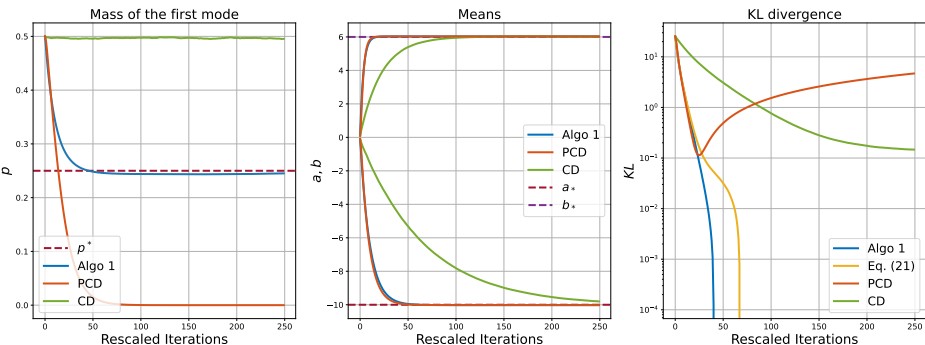

Figure 1: *GMM experiments:* Evolution of the parameters and the cross entropy during training by Algorithm 2, PCD, and CD. Average of 20 runs. *Left panels:* evolution of $p_k = 1/(1 + e^{-z_k})$; *middle panel:* evolution of $a_k$ and $b_k$; *right panel:* evolution of the Kullback-Leibler divergence. All three methods capture the location of the modes accurately, but only ours get the relative weights of these modes accurately (whereas PCD leads to mode collapse, and CD to an inaccurate estimate). Our method is also the only one that allows for direct estimation of the cross-entropy during training, and the only one performing GD on this cross-entropy–for better visualization we subtract the entropy of the target $H(\rho_*)$ and plot the Kullback-Leibler divergence instead of the cross-entropy.

In our numerical experiments, we set $d = 50$, use $N = 10^5$ walkers with a mini-batch of $N' = 10^4$ and $n = 10^5$ data points. We initialize the model at $\theta_0 = (a_0, b_0, z_0)$ with $a_0$ and $b_0$ drawn from an $N(0, \epsilon^2 I_d)$ with $\epsilon = 0.1$ and $z_0 = 0$, meaning that the initial $\rho_{\theta_0}$ is close to the PDF of an $N(0, I_d)$. The training is performed using Algorithm 2 with $h = 0.1$ and fixed learning rates $\gamma_k = 0.2$ for $a_k$ and $b_k$ and $\gamma_k = 1$ for $z_k$. We perform the resampling step by monitoring $ESS_k$ defined in (13) with constant $1/c_k = 1.05$ and using the systematic method. We also compare our results to those obtained using ULA with these same parameters (which is akin to training with the PCD algorithm) and with those obtained with the CD algorithm: in the latter case, we evolve the walkers by ULA with $h = 0.1$ for 4 steps between resets at the data points, and we adjust the learning rates by multiplying them by a factor 10. In all cases, we use the full batches of walkers, weights, and data points to estimate the empirical averages. We also use (12) to estimate the cross-entropy $H(\rho_{\theta_k}, \rho_*)$ during training by our method (CD and PCD do not provide estimates for these quantities), and in all cases compare the result with the estimate

$$\tilde{H}_k = \log\left((2\pi)^{d/2}\left(1 + e^{-z_*}\right)\right) - \frac{1}{n}\sum_{j=1}^{n}\log\left(e^{-\frac{1}{2}|x_*^j - a_k|^2} + e^{-\frac{1}{2}|x_*^j - b_k|^2 - z_k}\right) \qquad (16)$$

The results are shown in Figure 1. As can be seen, all three methods learn well the values of $a_*$ and $b_*$ specifying the positions of the modes. However, only our approach learns the value of $z$ specifying their relative weights. In contrast, the PCD algorithm leads to mode collapse, consistent with the theoretical explanation given in Appendix C.1, and the CD algorithm returns a biased value of $z$, consistent with the fact that it effectively uses the Fisher divergence as the objective. The results also show that the cross-entropy decreases with our approach, but bounces back up with the PCD algorithm and stalls with the CD algorithms: this is consistent with the fact that only our approach actually performs the GD on the cross-entropy, which, unlike the other algorithms, our approach estimates accurately during the training.

## 4.2 MNIST

Next, we perform empirical experiments on the MNIST dataset to answer the following question: when it comes to high-dimensional datasets with multiple modes, can our method produces an EBM that generates high-quality samples and captures the relative weights of the modes accurately?

To this end, we select a subset of MNIST consisting of only three digits: 2, 3, and 6. Then, we choose $5600$ images of label 2, $2800$ images of label 3, and $1400$ images of label 6 from the training set (for a total of $n = 9800$ data points), so that in this manufactured dataset the digits are in have respective weights $4/7$, $2/7$, and $1/7$.

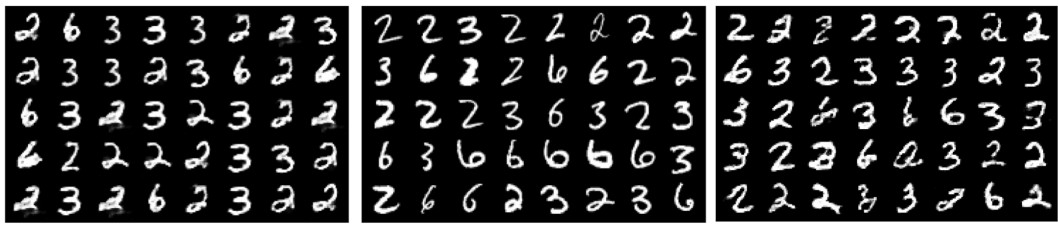

Generated samples (right after resampling)     Images from MNIST test set     Generated samples (without Jarzynski)

Figure 2: *MNIST: Left panel:* Examples of images generated by our method right after resampling in the last epoch. *Middle panel:* Images randomly selected from the test dataset of MNIST. *Right panel:* Examples of images generated by training using the persistent contrastive divergence (PCD) algorithm.

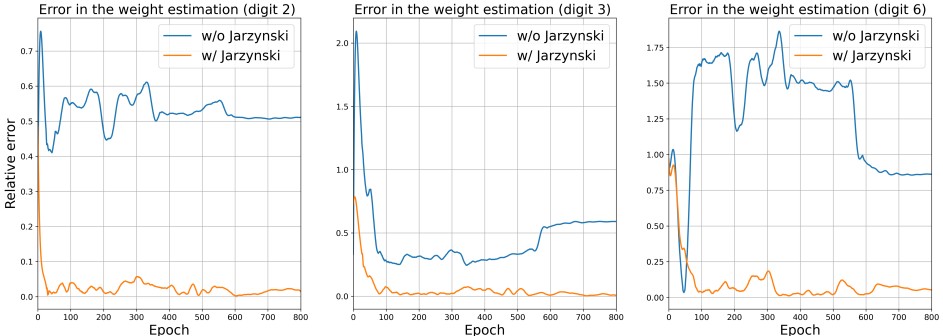

Figure 3: *MNIST:* Relative error of the weight estimation of the three modes (i.e. three digits). Our method outperforms the PCD algorithm in terms of recovery of the weight of each mode.

We train two EBMs to represent this data set, the first using our Algorithm 2, and the second using the PCD Algorithm 4. We represent the energy using a simple six-layer convolutional neural network with the swish activation and about $77K$ parameters. We use the ADAM optimizer for the training with a learning rate starting from $10^{-4}$ and linearly decaying to $10^{-10}$ until the final training step. The sample size of the walkers is set to $N = 1024$ and it is fixed throughout the training.

The results are shown in Figure 2. Both our Algorithm 1 and the PCD Algorithm 4 generate images of reasonable quality; as discussed in Appendix B.2, we observe that the Jarzynski weights can help us track the quality of generated images, and the resampling step is also helpful for improving this quality as well as the learned energy function.

However, the real difference between the methods comes when we look at the relative proportion of the digits generated in comparison to those in the data set. In Figure 3, we show the relative error on the weight estimation of each mode obtained by using a classifier pre-trained on the data set (see Appendix B.2 for details). Although the EBM trained with the PCD algorithm can discover all the modes and generate images of all three digits present in the training set, it cannot accurately recover the relative proportions of each digit. In contrast, our method is successful in both mode exploration and weight recovery.

Unlike in the numerical experiments done on Gaussian mixtures with teacher-student models, for the MNIST dataset, we cannot estimate the KL divergence throughout the training as we do not know the normalization constant of the true data distribution. Nevertheless, we can still use the formula (16) to track the cross-entropy between the data distribution and the walker distribution a plot of which is given in the right panel Figure 4 for the mini-batched training algorithm (Algorithm 2).

In these experiments with MNIST, we used the adaptive resampling scheme described after equation 13. In practice, we observed that few resamplings are needed during training, and they can often by avoided altogether if the learning rate is sufficiently small. For example, in the experiment reported in the left panel of Figure 4) a single resampling step was made. We also found empirically that the results are insensitive to the choice of of the parameters $c_k$ used for resampling: the rule of

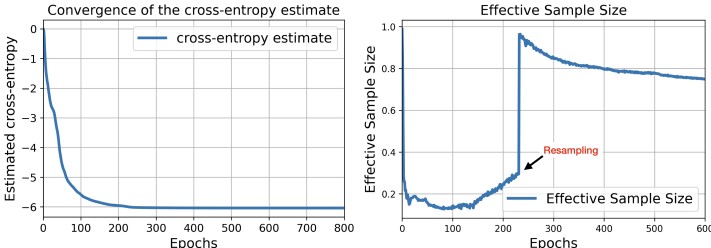

Figure 4: *MNIST dataset: Left Panel:* Convergence of the cross-entropy estimated by using formulae (12) with the mini-batched algorithm (Algorithm 2). *Right Panel:* Evolution of the effective sample size (ESS) defined in (13) – here resampling was started after 240 epochs (with $c_k = 0$ before and $c_k = 0.5$ afterwards), and occurred only once immediately after being switched on.

thumb we found is to not resample at the beginning of training, to avoid possible mode collapse, and use resampling towards the end, to improve the image quality.

### 4.3 CIFAR-10

We perform an empirical evaluation of our method on the full CIFAR-10 ($32 \times 32$) image dataset. We use the setup of [47], with the same neural architecture with $n_f = 128$ features, and compare the results obtained with our approach with mini-batching (Algorithm 2) to those obtained with PCD and PCD with data augmentation of [33] (which consists of a combination of color distortion, horizontal flip, rescaling, and Gaussian blur augmentations, to help the mixing of the MCMC sampling and stabilize training).

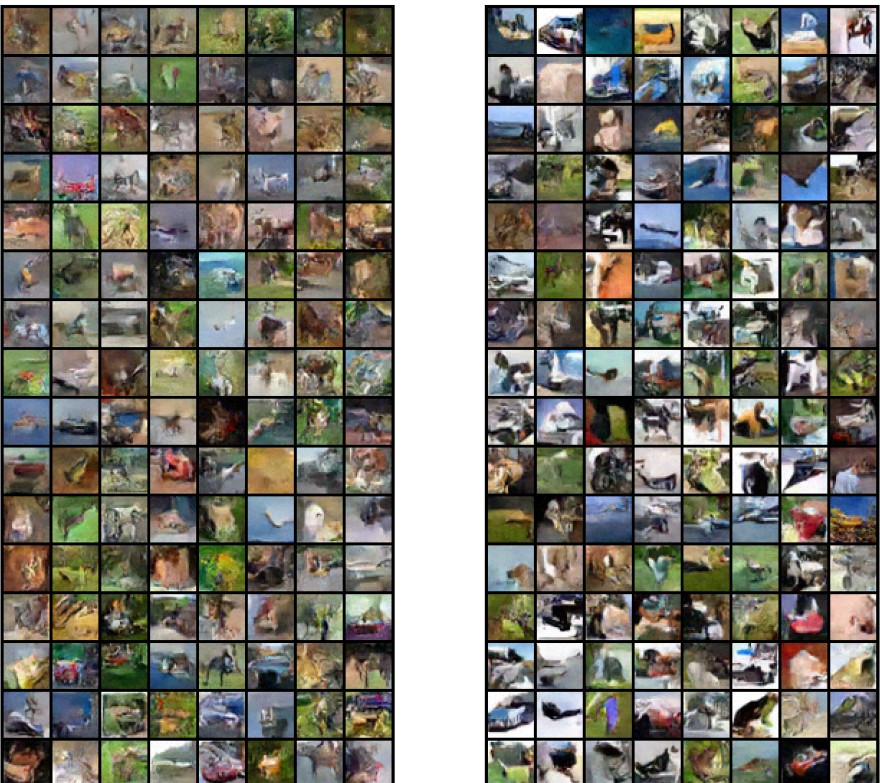

Generated CIFAR-10 samples with our approach   Generated CIFAR-10 samples with PCD

Figure 5: *CIFAR-10 dataset: Left panel*: images generated by training with Algorithm 2. *Right panel*: Images generated from the PCD with mini-batches. Up to date images available at `https://github.com/submissionx12/EBMs_Jarzynski`.

The code used to perform these new experiments is available in the anonymized GitHub referenced in our paper. The hyperparameters are the same in all cases: we take $N = 4096$ Langevin walkers with a mini-batch size $N' = 256$. We use the Adam optimizer with learning rate $\eta = 10^{-4}$ and inject a Gaussian noise of standard deviation $\sigma = 3 \times 10^{-2}$ to the dataset while performing gradient clipping in Langevin sampling for better performance. All the experiments were performed on a single A100 GPU. Training for 600 epochs took about 34 hours with the PCD algorithm (w/ and w/o data augmentation) and about 36 hours with our method.

Some of the images generated by our method are shown in Figure 5. We also quantitatively evaluate the performance of our models with the commonly used metrics (e.g. FID and Inception Score): the results are given in Table 1. They indicate that our method can achieve slightly better performance than the PCD algorithms w/ and w/o data augmentation at a similar computational cost. Furthermore, these results on CIFAR-10 suggest that our method scales well to complicated training tasks on more realistic data sets.

| Method | FID | Inception Score (IS) |
|---|---|---|
| PCD with mini-batches | 38.25 | 5.96 |
| PCD with mini-batches and data augmentation | 36.43 | 6.54 |
| Algorithm 2 with multinomial resampling | **32.18** | **6.88** |
| Algorithm 2 with systematic resampling | **30.24** | **6.97** |

Table 1: *textitCIFAR-10* dataset: Comparison of FID and Inception Score (IS) for PCD and Algorithm 2. Experiments performed using the neural architecture in [47] to model the energy.

It is worth stressing that a key component of the success of our method is the resampling step, as discussed in Appendix B.3 where we plot the Effective Sample Size (ESS) in Figure 13.

## 5   Concluding Remarks

In this work, we proposed a simple modification of the persistent contrastive divergence algorithm which corrects its biases and provably allows one to minimize the cross-entropy between the model and the target densities to train EBMs. Our approach rests on results from non-equilibrium thermodynamics (in particular, Jarzynski equality) that show how to take exact expectations over a probability density function (PDF) that is evolving in time—in the present context the PDF is that of the EBM, and it is changing as we train the model parameters. These formulas naturally lend themselves to practical implementation using sequential Monte Carlo sampling methods. The only difference with respect to training methods based on ULA is that the proposed approach maintains a set of weights associated with every walker. These weights correct for the bias induced by the evolution of the energy, and they also give a way to assess the quality of the samples generated by the method. On synthetic examples using Gaussian mixture densities as well as on some simple data sets involving MNIST, we showed that our method dodges some common drawbacks of other EBM training methods. In particular, it is able to learn the relative weights of various high-density regions in a multimodal distribution, thus ensuring a higher level of fairness than common EBM training techniques based on contrastive divergence.

To focus on the core idea, in the present study we put aside tuning considerations and did not sweep over the various hyperparameters and design choices, like learning rates, number of walkers, or resampling criteria and strategies. These considerations could greatly enhance the performance of our method. We believe that our results show that the method deserves deeper practical investigation on more realistic datasets and with more complex neural architectures.

## Acknowledgements

D.C. has worked under the auspices of Italian National Group of Mathematical Physics (GNFM). M.H. is supported by NYU McCracken doctoral fellowship. EVE is supported by the National Science Foundation under awards DMR-1420073, DMS-2012510, and DMS-2134216, by the Simons Collaboration on Wave Turbulence, Grant No. 617006, and by a Vannevar Bush Faculty Fellowship.

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

# A  Additional theoretical results

## A.1  Jarzynski equality in continuous-time

Before proving Proposition 1, we give a continuous-time version of this result whose proof helps guide the intuition.

**Proposition 2.** *Assume that the parameters $\theta$ are evolved according to some time-differentiable protocol $\theta(t)$ such that $\theta(0) = \theta_0$. Given any $\alpha > 0$, let $X_t \in \mathbb{R}^d$ and $A_t \in \mathbb{R}$ be the solutions of*

$$\begin{cases} dX_t = -\alpha \nabla U_{\theta(t)}(X_t)dt + \sqrt{2\alpha}\, dW_t, & X_0 \sim \rho_{\theta_0}, \\ \dot{A}_t = -\partial_\theta U_{\theta(t)}(X_t) \cdot \dot{\theta}(t), & A_0 = 0. \end{cases} \tag{17}$$

*where $U_\theta(x)$ is the model energy and $W_t \in \mathbb{R}^d$ is a standard Wiener process. Then, for any $t \geq 0$,*

$$\mathbb{E}_{\theta(t)}[\partial_\theta U_{\theta(t)}] = \frac{\mathbb{E}[\partial_\theta U_{\theta(t)}(X_t)e^{A_t}]}{\mathbb{E}[e^{A_t}]}, \qquad Z_{\theta(t)} = Z_{\theta_0}\mathbb{E}[e^{A_t}], \tag{18}$$

*where the expectations on the right-hand side are over the law of the joint process $(X_t, A_t)$.*

The second equation in (18) can also be written in term of the free energy $F_\theta = -\log Z_\theta$ as $F_{\theta(t)} = F_{\theta_0} - \log \mathbb{E}[e^{A_t}]$: this is Jarzynski's equality [34]. We stress that it is key to include the weights in (18) and, in particular, $\mathbb{E}[\partial_\theta U_{\theta(t)}(X_t)] \neq \mathbb{E}_{\theta(t)}[\partial_\theta U_{\theta(t)}]$. This is because the PDF of $X_t$ alone lags behind the model PDF $\rho_{\theta(t)}$: the larger $\alpha$, the smaller this lag, but it is always there if $\alpha < \infty$, see the remark at the end of this section for more discussion on this point. The inclusion of the weights in (18) corrects exactly for the bias induced by this lag.

An immediate consequence of Proposition 2 is that we can evolve $\theta(t)$ by the gradient descent flow over the cross-entropy by solving (17) concurrently with

$$\dot{\theta}(t) = \frac{\mathbb{E}[\partial_\theta U_{\theta(t)}(X_t)e^{A_t}]}{\mathbb{E}[e^{A_t}]} - \mathbb{E}_*[\partial_\theta U_{\theta(t)}], \qquad \theta(0) = \theta_0 \tag{19}$$

since by (18) the right hand side of (19) is precisely $-\partial_\theta H(\rho_{\theta(t)}, \rho_*) = \mathbb{E}_{\theta(t)}[\partial_\theta U_{\theta(t)}] - \mathbb{E}_*[\partial_\theta U_{\theta(t)}]$. Assuming that we know $Z_{\theta_0}$ we can also track the evolution of the cross-entropy (3) via

$$H(\rho_{\theta(t)}, \rho_*) = \log \mathbb{E}[e^{A_t}] + \log Z_{\theta_0} + \mathbb{E}_*[U_{\theta(t)}]. \tag{20}$$

*Proof.* The joint PDF $f(t, x, a)$ of the process $(X_t, A_t)$ satisfying (17) solves the Fokker-Planck equation (FPE)

$$\partial_t f = \alpha \nabla_x \cdot (\nabla_x U_{\theta(t)} f + \nabla_x f) + \partial_\theta U_{\theta(t)} \cdot \dot{\theta}(t)\partial_a f, \quad f(0, x, a) = Z_{\theta_0}^{-1}e^{-U_{\theta_0}(x)}\delta(a). \tag{21}$$

Let us derive an equation for

$$\hat{\rho}(t, x) = \int_{-\infty}^{\infty} e^a f(t, x, a)da \tag{22}$$

To this end, multiply (21) by $e^a$, integrate the result over $a \in (-\infty, \infty)$, and use integration by parts for the last term at the right-hand side to obtain:

$$\partial_t \hat{\rho} = \alpha \nabla_x \cdot (\nabla_x U_{\theta(t)}\hat{\rho} + \nabla_x \hat{\rho}) - \partial_\theta U_{\theta(t)} \cdot \dot{\theta}(t)\hat{\rho}, \qquad \hat{\rho}(0, x) = Z_{\theta_0}^{-1}e^{-U_{\theta_0}(x)} \tag{23}$$

By general results for the solutions of parabolic PDEs such as (23) (see [69], Chapter 7), we know that the solution to this equation is unique, and we can check by direct substitution that it is given by

$$\hat{\rho}(t, x) = Z_{\theta_0}^{-1}e^{-U_{\theta(t)}(x)}. \tag{24}$$

This implies that

$$\int_{\mathbb{R}^d} \hat{\rho}(t, x)dx = Z_{\theta_0}^{-1}Z_{\theta(t)}. \tag{25}$$

Since by definition $\mathbb{E}[e^{A_t}] = \int_{\mathbb{R}^d}\int_{-\infty}^{\infty} e^a f(t, x, a)dadx = \int_{\mathbb{R}^d}\hat{\rho}(t, x)dx$ this establishes the second equation in (18) . To establish the first notice that

$$\begin{aligned} \frac{\mathbb{E}[\partial_\theta U_{\theta(t)}(X_t)e^{A_t}]}{\mathbb{E}[e^{A_t}]} &= \frac{\int_{\mathbb{R}^d}\int_{-\infty}^{\infty}\partial_\theta U_{\theta(t)}(x)e^a f(t, x, a)dadx}{\int_{\mathbb{R}^d}\int_{-\infty}^{\infty}e^a f(t, x, a)dadx} = \frac{\int_{\mathbb{R}^d}\partial_\theta U_{\theta(t)}(x)\hat{\rho}(t, x)dx}{\int_{\mathbb{R}^d}\hat{\rho}(t, x)dx} \\ &= \frac{Z_{\theta_0}^{-1}\int_{\mathbb{R}^d}\partial_\theta U_{\theta(t)}(x)e^{-U_{\theta(t)}(x)}dx}{Z_{\theta_0}^{-1}Z_{\theta(t)}} = \mathbb{E}_{\theta(t)}[\partial_\theta U_{\theta(t)}] \end{aligned} \tag{26}$$

$\square$

**The need for Jarzynski's correction.** Suppose that the walkers satisfy the Langevin equation (first equation in (17)):

$$dX_t = -\alpha \nabla U_{\theta(t)}(X_t)dt + \sqrt{2\alpha}\,dW_t, \qquad X_0 \sim \rho_{\theta_0}, \tag{27}$$

where $\theta(t)$ is evolving according to some protocol. The probability density function $\rho(t, x)$ of $X_t$ then satisfies the Fokker-Planck equation (compare (23))

$$\partial_t \rho = \alpha \nabla \cdot \left( \nabla U_{\theta(t)}(x)\rho + \nabla \rho \right), \qquad \rho(t = 0) = \rho_{\theta_0} \tag{28}$$

The solution to this equation is not available in closed form, and in particular $\rho(t, x) \neq \rho_{\theta(t)}(x)$ – $\rho(t, x)$ is only close to $\rho_{\theta(t)}(x)$ if we let $\alpha \to \infty$, so that the walkers $X_t$ move much faster than the parameters $\theta(t)$ but this limit is not easily achievable in practice (as convergence of the FPE solution to its equilibrium is very slow in general if the potential $U_{\theta(t)}$ is complicated). As a result $\mathbb{E}[\partial_\theta U_{\theta(t)}(X_t)] \neq \mathbb{E}_{\theta(t)}[\partial_\theta U_{\theta(t)}]$, implying the necessity to include the weights in the expectation (18).

### A.2  Proof of Proposition 1

The iteration rule for $A_k$ in (5) implies that

$$A_k = \sum_{q=1}^{k} \left( \alpha_{q-1}(X_{q-1}, X_q) - \alpha_q(X_q, X_{q-1}) \right), \qquad k \in \mathbb{N}. \tag{29}$$

For $k \in \mathbb{N}_0$, let

$$\beta_k(x, y) = (4\pi h)^{-d/2} \exp\left( -\frac{1}{4h} |y - x + h\nabla U_{\theta_k}(x)|^2 \right) \tag{30}$$

be the transition probability density of the ULA update in (5), i.e. $\beta_k(X_k, X_{k+1})$ is the probability density of $X_{k+1}$ conditionally on $X_k$. By the definition of $A_k$, we have

$$\exp(A_k) = \prod_{q=1}^{k} \exp\left( \alpha_{q-1}(X_{q-1}, X_q) - \alpha_q(X_q, X_{q-1}) \right)$$
$$= e^{-U_{\theta_k}(X_k) + U_{\theta_0}(X_0)} \prod_{q=1}^{k} \frac{\beta_q(X_q, X_{q-1})}{\beta_{q-1}(X_{q-1}, X_q)} \tag{31}$$

where in the second line we added and subtracted $|X_q - X_{q-1}|^2/4h$ and used the definition of $\alpha_k(x, y)$ given in (6). Since the joint probability density function of the path $(X_0, X_1, \ldots, X_k)$ at any $k \in \mathbb{N}$ is

$$\varrho(x_0, x_1, \ldots, x_k) = Z_{\theta_0}^{-1} e^{-U_{\theta_0}(x_0)} \prod_{q=1}^{k} \beta_{q-1}(x_{q-1}, x_q) \tag{32}$$

we deduce from (31) and (32) that, given an $f : \mathbb{R}^d \to \mathbb{R}$, we can express the expectation $\mathbb{E}[f(X_k)e^{A_k}]$ as the integral

$$\mathbb{E}[f(X_k)e^{A_k}]$$
$$= \int_{\mathbb{R}^{dk}} f(x_k)e^{-U_{\theta_k}(x_k) + U_{\theta_0}(x_0)} \prod_{q=1}^{k} \frac{\beta_q(x_q, x_{q-1})}{\beta_{q-1}(x_{q-1}, x_q)} \varrho(x_0, x_1, \ldots, x_k)dx_0 \cdots dx_k$$
$$= Z_{\theta_0}^{-1} \int_{\mathbb{R}^{dk}} f(x_k)e^{-U_{\theta_k}(x_k)} \prod_{q=1}^{k} \beta_q(x_q, x_{q-1})dx_0 \cdots dx_k \tag{33}$$

Since $\int_{\mathbb{R}^d} \beta_k(x, y)dy = 1$ for all $k \in \mathbb{N}_0$ and all $x \in \mathbb{R}^d$, we can perform the integrals over $x_0$, then $x_1$, etc. in this expression to be left with

$$\mathbb{E}[f(X_k)e^{A_k}] = Z_{\theta_0}^{-1} \int_{\mathbb{R}^d} f(x_k)e^{-U_{\theta_k}(x_k)}dx_k \tag{34}$$

Setting $f(x) = 1$ in this expression gives

$$\mathbb{E}[e^{A_k}] = Z_{\theta_0}^{-1} \int_{\mathbb{R}^d} e^{-U_{\theta_k}(x_k)} dx_k = Z_{\theta_0}^{-1} Z_{\theta_k} \tag{35}$$

which implies the second equation in (7); setting $f(x_k) = \partial_\theta U_{\theta_k}(x_k)$ in (34) gives

$$\mathbb{E}[\partial_\theta U_{\theta_k}(X_k) e^{A_k}] = Z_{\theta_0}^{-1} \int_{\mathbb{R}^d} \partial_\theta U_{\theta_k}(x_k) e^{-U_{\theta_k}(x_k)} dx_k = Z_{\theta_0}^{-1} Z_{\theta_k} \mathbb{E}_{\theta_k}[\partial_\theta U_{\theta_k}] \tag{36}$$

which can be combined with (35) to arrive at the first equation in (7). $\qquad\square$

### A.3 Mini-batched version of Algorithm 1.

In Algorithm 2 we present a mini-batched version of Algorithm 1, where we do not update the positions of every walker at every iteration. Instead, we evolve only a small portion of the walkers, while keeping the other walkers frozen and only updating their weights using the information from the model update (which uses only the model energy and does not require back-propagation). This mini-batched version is more computationally efficient and leads to convergence in training with much fewer steps of ULA. More importantly, the mini-batched version of the algorithm, as compared to the full-batched version, enlarges the total sample size and therefore improves the sample variety with even less computational cost.

---

**Algorithm 2** Mini-batched Sequential Monte-Carlo training with Jarzynski correction

---

1: **Inputs:** data points $\{x_i^*\}_{i=1}^n$; energy model $U_\theta$; optimizer step $\mathrm{opt}(\theta, \mathcal{D})$ using $\theta$ and the empirical CE gradient $\mathcal{D}$; initial parameters $\theta_0$; number of walkers $N \in \mathbb{N}_0$; batch size $N' \in \mathbb{N}_0$ with $N' < N$, set of walkers $\{X_0^i\}_{i=1}^N$ sampled from $\rho_{\theta_0}$; total duration $K \in \mathbb{N}$; ULA time step $h$; set of positive constants $\{c_k\}_{k \in \mathbb{N}}$.

2: $A_0^i = 0$ for $i = 1, \dots, N$.

3: **for** $k = 1 : K - 1$ **do**

4: $\quad p_k^i = \exp(A_k^i) / \sum_{j=1}^N \exp(A_k^j)$ $\hfill \triangleright$ normalized weights

5: $\quad \tilde{\mathcal{D}}_k = \sum_{i=1}^N p_k^i \partial_\theta U_{\theta_k}(X_k^i) - n^{-1} \sum_{j=1}^n \partial_\theta U_{\theta_k}(x_*^j)$ $\hfill \triangleright$ empirical CE gradient

6: $\quad \theta_{k+1} = \mathrm{opt}(\theta_k, \tilde{\mathcal{D}}_k)$ $\hfill \triangleright$ optimization step

7: $\quad$ Randomly select a mini-batch $\{X_k^j\}_{j \in B}$ with $\#B = N'$ from the set of walkers $\{X_k^i\}_{i=1}^N$

8: $\quad$ **for** $j \in B$ **do**

9: $\qquad X_{k+1}^j = X_k^j - h\nabla U_{\theta_k}(X_k^j) + \sqrt{2h}\, \xi_k^j, \qquad \xi_k^j \sim \mathcal{N}(0_d, I_d)$ $\hfill \triangleright$ ULA

10: $\qquad A_{k+1}^j = A_k^j - \alpha_{k+1}(X_{k+1}^j, X_k^j) + \alpha_k(X_k^j, X_{k+1}^j)$ $\hfill \triangleright$ weight update

11: $\quad$ **end for**

12: $\quad$ **for** $j \notin B$ **do**

13: $\qquad X_{k+1}^j = X_k^j$ $\hfill \triangleright$ no update of the walkers

14: $\qquad A_{k+1}^j = A_k^j - U_{\theta_{k+1}}(X_k^j) + U_{\theta_k}(X_k^j)$ $\hfill \triangleright$ weight update

15: $\quad$ **end for**

16: $\quad$ Resample the walkers and reset the weights if $\mathrm{ESS}_{k+1} < c_{k+1}$, see (13). $\triangleright$ resampling step

17: **end for**

18: **Outputs:** Optimized energy $U_{\theta_K}$; set of weighted walkers $\{X_K^i, A_K^i\}_{i=1}^N$ sampling $\rho_{\theta_K}$; partition function estimate $\tilde{Z}_{\theta_K} = Z_{\theta_0} N^{-1} \sum_{i=1}^N \exp(A_K^i)$; CE estimate $\log \tilde{Z}_{\theta_K} + n^{-1} \sum_{j=1}^n U_{\theta_K}(x_*^j)$

---

### A.4 Resampling Routines

Resampling schemes are necessary to tackle the decay of effective sample size (13). For the sake of completeness, here we recall three of the most widely used routines: multinomial [70], stratified [71], and systematic resampling [72], and refer the reader to the review [73] for more details.

Given a set of normalized scalar weights $\{p_i\}_{i=1}^N \in [0, 1]$ with $\sum_{i=1}^N p_i = 1$ associated to the $N$ walkers, we define the cumulative sum

$$P_n = \sum_{i=1}^n p_i, \qquad n = 1, \dots, N \tag{37}$$

All three methods prescribe a way to choose a set $\{u_n\}_{n=1}^N \in (0,1]$ used to perform the resampling in the following way: for every $n, m \in \{1, \ldots, N\}$, the $m$-th particle is chosen during the $n$-th extraction if

$$P_{m-1} < u_n < P_m \tag{38}$$

Let us now specify how the set of $u_n$ is selected in the cases in study. We denote by $\mathcal{U}(a,b]$ the uniform probability distribution on the interval $(a, b]$.

**Multinomial resampling.** Sample $u_n^{\text{mult}} \sim \mathcal{U}(0,1]$, independently for every $n \in \{1, \ldots, N\}$. This approach leads to a large number of possible resampled configurations, which is not desirable in practice as it increases the variance of the estimator.

**Stratified resampling.** Partition the interval $(0,1]$ into $N$ sub-intervals, or strata, of size $1/N$; then, sample $u_n^{\text{str}} \sim \mathcal{U}((n-1)/N, n/N]$ independently for each $n = \{1, \ldots, N\}$. This approach picks a single $u_n$ in each stratus, thereby reducing the number of possible resampled configurations.

**Systematic resampling.** Partition the interval $(0,1]$ into $N$ sub-intervals, or strata, of size $1/N$; then, sample $u_1^{\text{sys}} \sim \mathcal{U}(0, 1/N)$, and $u_n^{\text{sys}} = u_1^{\text{sys}} + (n-1)/N$ for $n > 1$. This method also reduces the number of possible resampled configurations.

Note that there are various modifications of these three methods (see the review [73]), all of them meeting the so-called unbiasedness condition, that is, the $i$-particle is expected to be sampled in average $N p_i$ times. However, these extensions do not lead to a critical lowering of the number of possible resampled configurations compared to systematic resampling. For this reason in our numerical experimentals we only tested the three methods above. Note also that, regardless of the resampling routine one uses, a fundamental role is played by the variance of the weights: if the cumulative sum is dominated by one or very few weights, i.e. resampling is triggered too late, it does not remedy the suppression of population variability.

### A.5 Contrastive divergence and persistent contrastive divergence algorithms

For completeness, we give the CD and PCD algorithms we used to compare with our Algorithms 1 and 2. As written these algorithms use the full batch of data points at every optimization step, but they can easily be modified to do mini-batching.

---
**Algorithm 3** Contrastive divergence (CD) algorithm

---
1: **Inputs:** data points $\Omega = \{x_*^i\}_{i=1}^n$ in $\mathbb{R}^d$; energy model $U_\theta$; optimizer step $\text{opt}(\theta, \mathcal{D})$ using $\theta$ and the empirical gradient $\mathcal{D}$; initial parameters $\theta_0$; number of walkers $N \in \mathbb{N}_0$ with $N < n$; total duration $K \in \mathbb{N}$; ULA time step $h$; $P \in \mathbb{N}$.
2: **for** $k = 1, \ldots, K-1$ **do**
3:     **for** $i = 1, ..., N$ **do**
4:         $X_0^i = SampleMultinomial(\Omega)$
5:         **for** $p = 0, ..., P-1$ **do**
6:             $X_{p+1}^i = X_p^i - h \nabla U_{\theta_k}(X_p^i) + \sqrt{2h}\,\xi_p^i, \qquad \xi_p^i \sim \mathcal{N}(0_d, I_d)$       ▷ ULA
7:         **end for**
8:     **end for**
9:     $\tilde{\mathcal{D}}_k = N^{-1} \sum_{i=1}^N \partial_\theta U_{\theta_k}(X_P^i) - n^{-1} \sum_{i=1}^n \partial_\theta U_{\theta_k}(x_*^i)$       ▷ empirical gradient
10:     $\theta_{k+1} = \text{opt}(\theta_k, \tilde{\mathcal{D}}_k)$       ▷ optimization step
11: **end for**
12: **Outputs:** Optimized energy $U_{\theta_K}$; set of walkers $\{X_P^i\}_{i=1}^N$

---

Interestingly, we can write down an equation that mimics the evolution of the PDF of the walkers in the CD algorithm, at least in the continuous-time limit: this equation reads

$$\partial_t \check{\rho} = \alpha \nabla \cdot \left( \nabla U_{\theta(t)}(x)\check{\rho} + \nabla\check{\rho} \right) - \nu(\check{\rho} - \rho_*), \qquad \check{\rho}(t=0) = \rho_* \tag{39}$$

where the parameter $\nu > 0$ controls the rate at which the walkers are reinitialized at the data points: the last term in (39) is a birth-death term that captures the effect of these reinitializations. The solution

**Algorithm 4** Persistent contrastive divergence (PCD) algorithm

---

1: **Inputs:** data points $\Omega = \{x_i^*\}_{i=1}^n$ in $\mathbb{R}^d$; energy model $U_\theta$; optimizer step $\mathrm{opt}(\theta, \mathcal{D})$ using $\theta$ and the empirical CE gradient $\mathcal{D}$; initial parameters $\theta_0$; number of walkers $N \in \mathbb{N}_0$ with $N < n$; total duration $K \in \mathbb{N}$; ULA time step $h$.
2: $X_0^i = SampleMultinomial(\Omega)$ for $i = 1, \ldots, N$.
3: **for** $k = 1, \ldots, K - 1$ **do**
4:     $\tilde{\mathcal{D}}_k = N^{-1} \sum_{i=1}^N \partial_\theta U_{\theta_k}(X_k^i) - n^{-1} \sum_{i=1}^n \partial_\theta U_{\theta_k}(x_*^i)$         ▷ empirical gradient
5:     $\theta_{k+1} = \mathrm{opt}(\theta_k, \tilde{\mathcal{D}}_k)$         ▷ optimization step
6:     **for** $i = 1, \ldots, N$ **do**
7:         $X_{k+1}^i = X_k^i - h\nabla U_{\theta_k}(X_k^i) + \sqrt{2h}\,\xi_k^i, \qquad \xi_k^i \sim \mathcal{N}(0_d, I_d)$         ▷ ULA
8:     **end for**
9: **end for**
10: **Outputs:** Optimized energy $U_{\theta_K}$; set of walkers $\{X_K^i\}_{i=1}^N$.

---

to this equation is not available in closed from (and $\check{\rho}(t, x) \neq \rho_{\theta(t)}(x)$ in general), but in the limit of large $\nu$ (i.e. with very frequent reinitializations), we can show [55] that

$$\check{\rho}(t, x) = \rho_*(x) + \nu^{-1}\alpha\nabla \cdot \left(\nabla U_{\theta(t)}(x)\rho_*(x) + \nabla\rho_*(x)\right) + O(\nu^{-2}). \tag{40}$$

As a result

$$\begin{aligned}
&\int_{\mathbb{R}^d} \partial_\theta U_{\theta(t)}(x)(\rho_*(x) - \check{\rho}(t, x))dx \\
&= -\nu^{-1}\int_{\mathbb{R}^d} \partial_\theta U_{\theta(t)}(x)\nabla \cdot \left(U_{\theta(t)}(x)\rho_*(x) + \nabla\rho_*(x)\right)dx + O(\nu^{-2}) \\
&= \nu^{-1}\int_{\mathbb{R}^d} \left(\partial_\theta\nabla U_{\theta(t)}(x) \cdot \nabla U_{\theta(t)}(x) - \partial_\theta\Delta U_{\theta(t)}(x)\right)\rho_*(x)dx + O(\nu^{-2})
\end{aligned} \tag{41}$$

The leading order term at the right hand side is precisely $\nu^{-1}$ times the gradient with respect to $\theta$ of the Fisher divergence

$$\begin{aligned}
&\frac{1}{2}\int_{\mathbb{R}^d} |\nabla U_\theta(x) + \nabla\log\rho_*(x)|^2\rho_*(x)dx \\
&= \frac{1}{2}\int_{\mathbb{R}^d} \left[|\nabla U_\theta(x)|^2 - 2\Delta U_\theta(x) + |\nabla\log\rho_*(x)|^2\right]\rho_*(x)dx
\end{aligned} \tag{42}$$

where $\Delta$ denotes the Laplacian and we used $\int_{\mathbb{R}^d} \nabla U_\theta(x) \cdot \nabla\log\rho_*(x)\rho_*(x)dx = \int_{\mathbb{R}^d} \nabla U_\theta(x) \cdot \nabla\rho_*(x)dx = -\int_{\mathbb{R}^d} \Delta U_\theta(x)\rho_*(x)dx$. This confirms the known fact that the CD algorithm effectively performs GD on the Fisher divergence rather than the cross-entropy [28].

# B    Additional numerical results

Here we give additional details about our numerical results. The codes are available at: `https://github.com/submissionx12/EBMs_Jarzynski`.

## B.1    Gaussian mixture distributions

Here we give some additional numerical results for the teacher-student model discussed in Section 4.1 in which the two wells of the teacher are aligned along the first dimension, fixing the first component of the means to be $a_*^1 = -10$ and $b_*^1 = 6$, and $a_*^\alpha = b_*^\alpha = 0$ for any $\alpha = 2, \ldots, d$; we also set $z_* = -\log(3)$, corresponding to a mass $p_* = 1/(1 + e^{-z_*}) = 0.25$ of the mode centered at $a$.

All the simulations are performed in $d = 50$, with a time step of $h = 0.1$ for the ULA update. The number of data points is $n = 10^4$. The setup of the teacher is the same for every simulation we display here and the optimization step is performed with full batch gradient descent with learning rate constant in time.

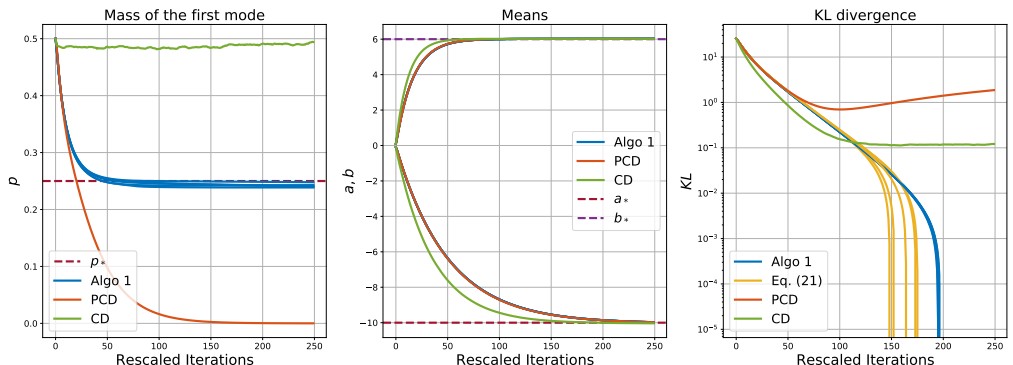

Figure 6: *GMM experiments:* Evolution of the parameters and the cross entropy during training by Algorithm 1, PCD, and CD. W.r.t. Algorithm 1, we display the results for five different thresholds in the resampling step. *Left panels:* evolution of $p_k = 1/(1 + e^{-z_k})$; *middle panel:* evolution of $a_k$ and $b_k$; *right panel:* evolution of the Kullback-Leibler divergence.

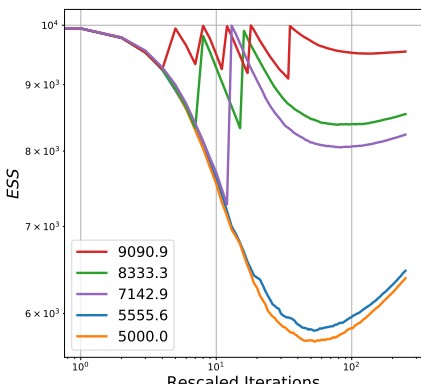

Figure 7: *GMM experiments:* Evolution of the effective sample size (ESS) for five different choices of threshold associated to $c$, constant in time. Bimodal student.

**GMM for different choices of** $c_k$. Results are shown in Figures 6 and 7. As initial conditions we select $a_0^1 = -10^{-1}$, $b_0^1 = 10^{-1}$, $a_0^\alpha \sim 10^{-2}\mathcal{N}(0,1)$ and $b_0^\alpha \sim 10^{-2}\mathcal{N}(0,1)$ for any $\alpha = 2, \ldots, d$; this perturbation around $a = b = 0$ is prescribed to avoid numerical degeneracy. For $z$, we fix $z_0 = 0$. We run Algorithm 1, as well as the CD and PCD Algorithms 3 and 4 using $N = 10^4$ walkers for $K = 8 \cdot 10^3$ iterations. We use a different learning rate for $z$ and $a, b$, namely $\gamma_z = 0.125$ and $\gamma_{a,b} = 0.2\gamma_z$. Moreover, these values are multiplied by a factor $10$ in CD. With regard to the resampling step in Algorithm 1, we choose a threshold $c_k = c$ which is fixed in time. We display the result for five possible values of this hyperparameter, namely $c = [0.1, 0.2, 0.4, 0.8, 1.0]$; these are related to thresholds in the effective sampling size (ESS) via $\text{ESS}_{\text{thresh}} = N/(c+1)$. With regard to Algorithm 1, we choose $M = 1$ and $N' = N$, that is the full batch version; in the CD Algorithm 4 we choose $P = 4$ for the number of ULA steps between restarts. Every run is performed five times and the average between them is shown in figures.

**Mode collapse in GMM for PCD.** Results are shown in Figure 8. In the same setup as above we select as initial conditions $a_0^1 = -10^{-1}$, $b_0^1 = 10^{-1}$, $a_0^\alpha \sim 10^{-2}\mathcal{N}(0,1)$ and $b_0^\alpha \sim 10^{-2}\mathcal{N}(0,1)$ for any $\alpha \in [2, d]$; for $z$, we fix $z_0 = 0$. The learning rate is chosen to be $\gamma_z = 5$ for $z$ and $\gamma_{a,b} = 0.2\gamma_z$ for the means. The time step of ULA is $h = 0.2$. Since the objective is to show mode collapse in the PCD algorithm, we run just Algorithm 4 for $K = 10^4$ iterations and $N = 10^4$ walkers.

**The need for resampling.** In absence of resampling, we observe a dramatic deterioration of ESS (Figure 9). To investigate how this issue is solved by resampling, we compare the three routines presented in Appendix A.4, using three pre-specified lags between the resampling steps. We use the same hyperparameters (learning rate, target distribution, etc.) as in Figure 9. The results are

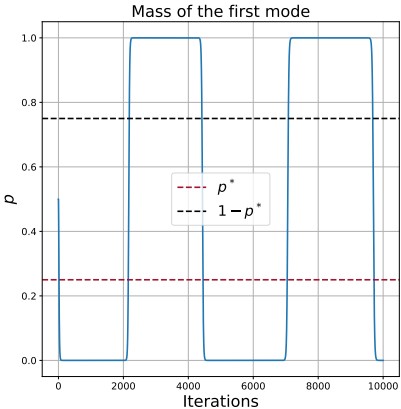

Figure 8: Mode collapse and oscillations in PCD. Evolution of the probability $p_k = 1/(1 + e^{-z_k})$.

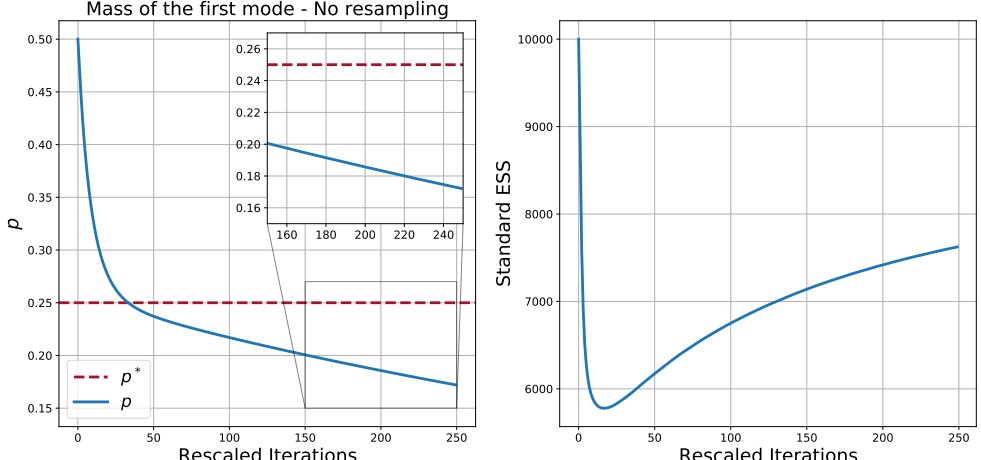

Figure 9: GMM in 50d without resampling step, full batch experiment with $N = 10^5$ walkers, average of 30 runs, $4.8 \cdot 10^5$ iterations, same target density of Subsection 4.1. *Left Panel:* relative mass of the first mode. *Right panel:* evolution of ESS

shown in Figure 10: looking at the upper panels, we see that stratified and systematic resampling are more stable than multinomial; moreover, if the resampling step is performed too infrequently (green lines), the method converges to a result where the target relative mass is off. Looking at the lower panels, we see that the minimum statistical error is obtained with systematic resampling; moreover, the behaviour of uncertainty appears to be more stable than stratified.

As a side note, systematic resampling requires just a single random number, contrarily to multinomial and stratified which need $N$, making the former also more efficient from a computational point of view. This consideration, plus the experimental results we just discussed, strongly motivate the adoption of such method for the resampling step in our proposed algorithm.

**Discussion.** GMM in high dimension are challenging for the standard CD and PCD algorithms given in 3 and 4. The experimental results of this section also confirm the theoretical analysis prsented in Appendix C below: CD is performing GD but on Fisher divergence rather than cross entropy, and as a result it incorrectly estimates the mass of the modes since Fisher divergence is insensitive to this quantity. On the other hand, PCD causes cycles of mode collapse (see Figure 8 and the left panel of Figure 6), and the KL divergence does not decreases monotonically, since the protocol is not ensured to be gradient descent (right panel in Figure 6).

In this example, our Algorithm 1 outperforms these standard methods as it is an implementation GD on cross-entropy. In particular, the estimation of the relative mass with Algorithm 1 is more

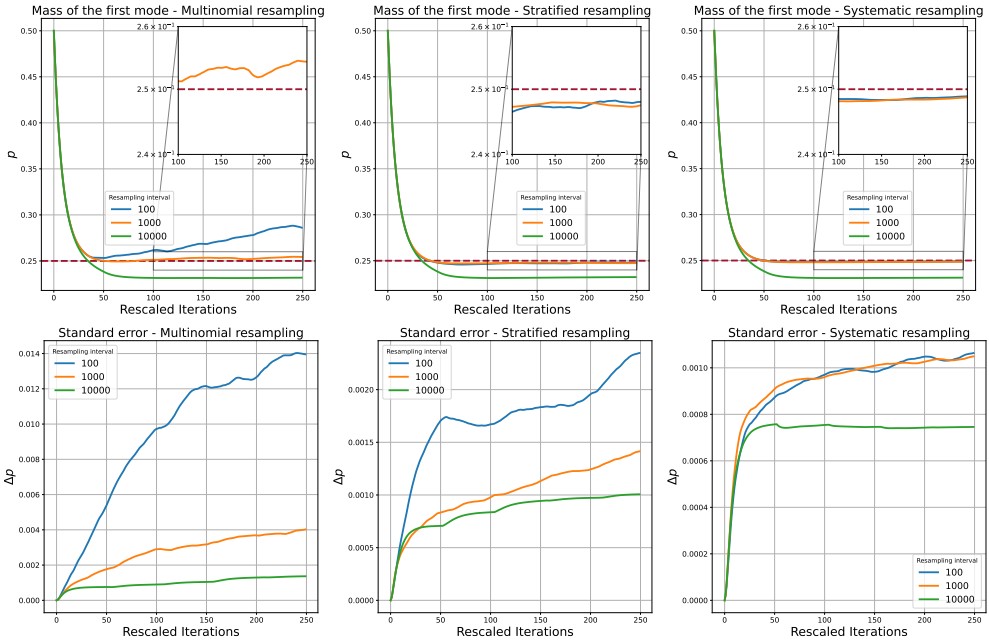

Figure 10: GMM in 50d with three resampling routines, full batch experiment with $N = 10^5$ walkers, $50$ runs per each resampling interval, $4.8 \cdot 10^5$ iterations. *Upper panels:* relative mass of the first mode for each resampling method. *Lower panels:* standard error computed using the empirical standard deviation via $\sigma_{emp}/\sqrt{50}$.

accurate than with the CD and PCD algorithms (see Figure 6). Moreover, the computation of KL divergence via Jarzynski weights is fairly precise; in Figure 6 the estimated KL and the exact one overlaps beyond the minimum values reached in with the CD and PCD algorithms. With regard to Figure 7, the choice of the threshold for resampling does not appear to be decisive in this regime of hyperparameters; in fact, looking at the evolution of $\theta$ and of KL divergence, the overall behavior of Algorithm 1 is not dramatically influenced by the choice of $c$.

### B.2 MNIST data set

In this section, we provide additional details about our numerical experiments on the MNIST dataset using Algorithm 2. First, as discussed in Section 4.2, we confirm that the Jarzynski weights of the generated samples are directly related to the image quality: this is shown in the left panel of Figure 11, where we display images along with their Jarzynski weights. Moreover, the right panel of Figure 11 indicates that resampling at the end of training using these weights helps improve sample quality. Examples of generated images are shown in Figure 12.

### B.3 CIFAR-10

A key component of the success of our method is the resampling step. In Figure 13 we plot the Effective Sample Size (ESS): each peak in Figure 13 indicates a step of resampling of all the samples. We note that training EBMs with mini-batches has the side effect of a rapid loss of the ESS, which measures the sample quality. So, doing resampling with a proper criterion and a reliable resampler is necessary.

## C  Theoretical analysis of mode collapse in simple 1d-GMM

In this appendix, we explain why and how mode collapse arises when learning multimodal distributions if we do not include the weights prescribed by Jarzynski equality, and why collapse does not arise with these weights included. To show this in the simplest setting, we work with a Gaussian mixture model: a mixture of two one-dimensional Gaussian densities, with the same variance equal

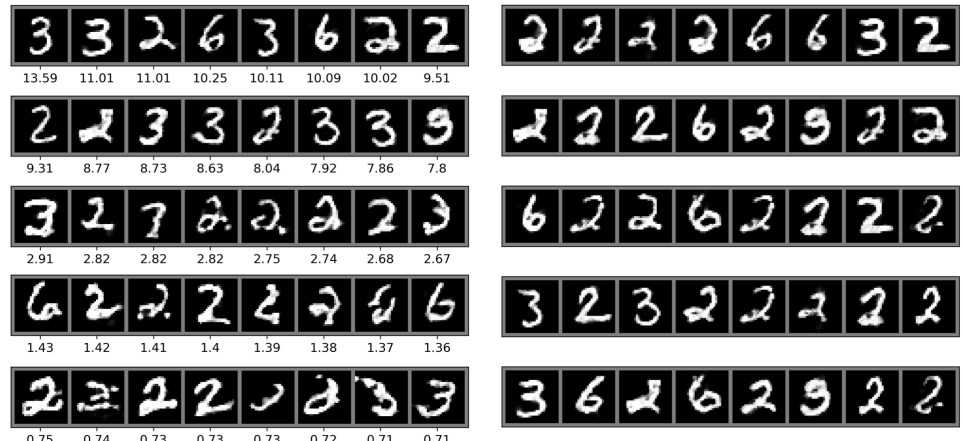

Generated samples during training (w/o resampling)     Generated samples during training (w/ resampling)

Figure 11: *MNIST dataset: Left panel:* Images randomly chosen during the training from the entire set of generated samples with their associate Jarzynski weights. From left to right, top to bottom, the higher the Jarzynski weight is, the better the image quality is. *Right panel:* Images obtained under the same training conditions, after resampling and continued training for 120 epochs.

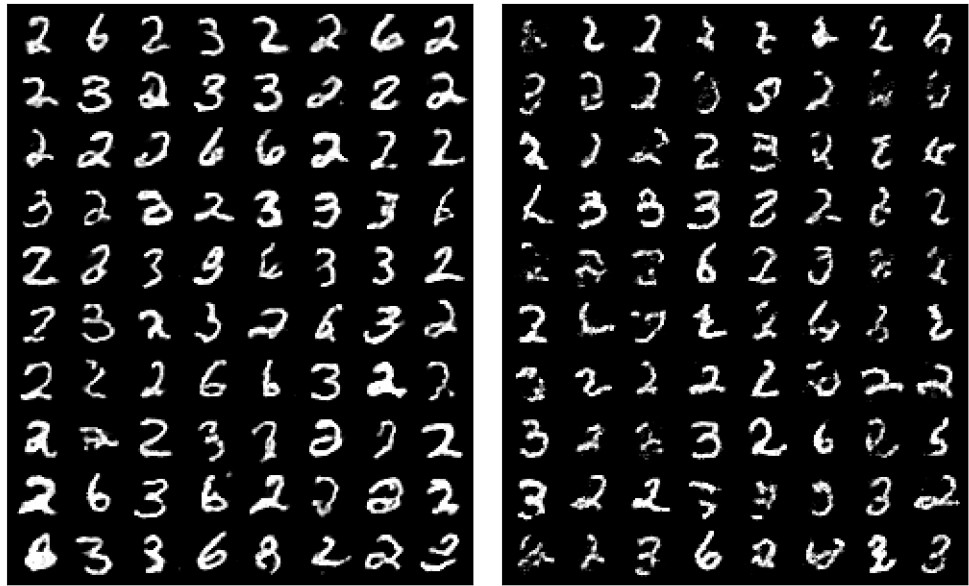

Figure 12: *MNIST dataset: Left panel*: images generated by the mini-batched version of the Algorithm 2. *Right panel*: Images generated from the PCD with mini-batches.

to 1, with means $a, b \in \mathbb{R}$ of the modes: our aim is to learn the probability masses of these modes. The general picture is given in Appendix C.1, with details presented afterwards in Appendix C.2, where we assume that the learning dynamics is that in (47), with no walkers and direct access to the distributions $\rho_{z(t)}$ and $\rho_*$, and Appendix C.3, where we quantify the stochastic fluctuations induced by empirical estimation.

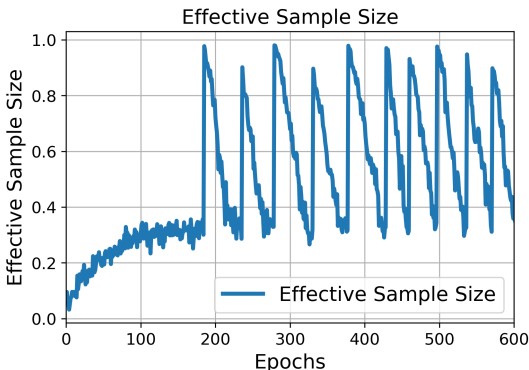

Figure 13: *CIFAR-10 dataset:* The Effective Sample Size (ESS) during the training with Algorithm 2. Each peak in the plot implies one step of resampling. Notice that the number of resampling in CIFAR-10 experiments is significantly larger than the one in MNIST experiments ( 4), which suggests the necessity of resampling in the scalability of our method.

## C.1    Mode collapse in Gaussian Mixtures

### C.1.1    Target distribution and model parameterization

As target density we take
$$\rho_*(x) = \frac{e^{-\frac{1}{2}|x-a|^2} + e^{-z_* - \frac{1}{2}|x-b|^2}}{\sqrt{2\pi}(1 + e^{-z_*})}. \tag{43}$$

Here $z_* \in \mathbb{R}$ parameterizes the mass of the second mode relative to the first and is the sole parameter of interest. The proportion of samples in both modes is indeed
$$p_* := \frac{1}{1 + e^{-z_*}}, \qquad q_* := 1 - p_* = \frac{e^{-z_*}}{1 + e^{-z_*}}. \tag{44}$$

Our point is that when both modes are separated by very low-density regions, learning $z_*$ without weight correction leads to an incorrect estimation of the mode probabilities ('no-learning') or to mode collapse depending on the initialization of the learning procedure, whereas using the Jarzynski correction does not. From now on, we will suppose that the modes are separated in the following sense:
$$|a - b| = 10, \tag{A1}$$

which will be enough for our needs. The more separated they are, the stronger our quantitative bounds will be.

The parameterization for our model potential $U_z$ is consistent with (43):
$$U_z(x) = -\log\left(e^{-\frac{1}{2}|x-a|^2} + e^{-z - \frac{1}{2}|x-b|^2}\right) \tag{45}$$

and the associated partition function and free energy are
$$Z_z = \sqrt{2\pi}\left(1 + e^{-z}\right), \qquad F_z = -\log Z_z = -\log(1 + e^{-z}) - \frac{1}{2}\log(2\pi). \tag{46}$$

The normalized probability density associated with $U_z(x)$ is thus $\rho_z(x) = e^{-U_z(x) + F_z}$.

### C.1.2    Learning procedures

Gradient descent on the cross entropy leads to the following continuous-time dynamics:
$$\dot{z}(t) = \mathbb{E}_{z(t)}[\partial_z U_{z(t)}] - \mathbb{E}_*[\partial_z U_{z(t)}]. \tag{47}$$

For simplicity, we will start this ODE at $z(0) = 0$. It corresponds to a proportion of $\frac{1}{2}$ for both modes.

The gradient descent (47) is an ideal situation where the expectations $\mathbb{E}_{z(t)}, \mathbb{E}_*$ can be exactly analyzed. In practice however, the two terms of (47) are estimated; the second term using a finite

number of training data $\{x_*^i\}_{i=1}^n$, and the first one using a finite number of walkers $\{X_t^i\}_{i=1}^N$, with associated weights $\{e^{A_t^i}\}_{i=1}^N$. For simplicity we set $N = n$ and the empirical GD dynamics is thus

$$\dot{z}(t) = \frac{\sum_{i=1}^n e^{A_t^i} \partial_z U_{z(t)}(X_t^i)}{\sum_{i=1}^n e^{A_t^i}} - \frac{\sum_{i=1}^n \partial_z U_{z(t)}(x_*^i)}{n} \tag{48}$$

and the walkers evolve under the Langevin dynamic

$$dX_t^i = -\alpha \nabla U_{z(t)}(X_t^i)dt + \sqrt{2\alpha}dW_t^i \tag{49}$$

for some fixed $\alpha > 0$. Now the nature of the algorithm varies depending on how the walkers are initialized and the Jarzynski weights are evolved.

1. The standard PCD algorithm sets $X_0^i = x_*^i$, that is, the walkers are initialized at the data points, and the weights are not evolved, that is $A_t^i = 0$ at any time.

2. Alternatively, the walkers could be initialized at samples of the initial model: $X_0^i \sim \rho_{z(0)}$, with the weights not evolved. We refer to this algorithm as the *umweighted* procedure.

3. Our algorithm 2 corresponds to initializing the walkers at samples of the initial model, $X_0^i \sim \rho_{z(0)}$, and uses the Jarzynski rule (17) for the weights updates.

For simplicity we analyze the outcome of these algorithms in the continuous-time set-up, i.e. using (48). This is an idealization of the actual algorithms, but it makes the analysis more transparent.

### C.1.3   Perturbative analysis

Using simple approximations (see Appendix C for details) , we show that in the three cases above, the dynamics (48) can be seen as a perturbation of a simpler differential system whose qualitative behaviour fits with our numerical simulations. These systems depend on the initialization of the walkers and are thus prone to small stochastic fluctuations. We introduce:

- $\hat{q}_*$ the proportion of training data $\{x_*^i\}$ that are close to mode $b$ (a more precise definition is given in Appendix C), and we define $\hat{z}_*$ as satisfying $\hat{q}_* = e^{-\hat{z}_*}/(1 + e^{-\hat{z}_*})$;

- $\hat{q}(0)$ the proportion of walkers at initialization that are close to $b$, and $\hat{p}(0) = 1 - \hat{q}(0)$.

Practically, $\hat{q}_*$ is a random variable centered at $q_*$ and with fluctuations of order $n^{-1/2}$, and $\hat{q}(0)$ is centered at $e^{-z(0)}/(1+e^{-z(0)}) = \frac{1}{2}$ with fluctuations of the same order. In the limit where $n$ is large, they can be neglected, but in more realistic training settings, the use of mini-batches leads to small but non-negligible fluctuations, as will be clear in Equation (53).

The arguments in Appendix C lead to the following approximations:

- In the model-initialized algorithm without Jarzynski correction ('unweighted'), (48) is a perturbation of

$$\dot{z}_{\text{unw}}(t) = \hat{q}(0) - \hat{q}_*. \tag{50}$$

This system has no stable fixed point since the RHS no longer depends on $z_{\text{unw}}(t)$, leading to a linear drift $z_{\text{unw}}(t) = (\hat{q}(0) - \hat{q}_*)t$ and thus to a divergence of $z_{\text{unw}}(t)$. Consequently, the mass of the second mode, $q(t) = e^{-z_{\text{unw}}(t)}/(1 + e^{-z_{\text{unw}}(t)})$, converges to 0 or 1. However, on longer time scales, this drift leads to a sudden transfer of all walkers in one the modes, then to a complete reversal of the drift of $z_{\text{unw}}$ which then diverges in the opposite direction, leading to a succession of alternating mode-collapses; see also Figure 8 and Remark C.4 below.

- In the continuous-time version of the standard PCD algorithm, (48) is a perturbation of the same ODE as (50). However, in this context, since the initial data $\{X_0^i\}$ and the training data $\{x_*^i\}$ are identical, we have $\hat{q}(0) = \hat{q}_*$, and

$$\dot{z}_{\text{pcd}}(t) = 0. \tag{51}$$

The parameters do not evolve and the system is stuck at $z_{\text{pcd}}(0)$ ('no-learning'). Note however that in this version of PCD, the walkers are initialized at the *full* training data. In practice, the number of walkers is smaller than the number of training data so that one often uses a small batch of training data to initialize them; in this case we can still have $\hat{q}(0) \neq \hat{q}_*$, falling back to the first case above.

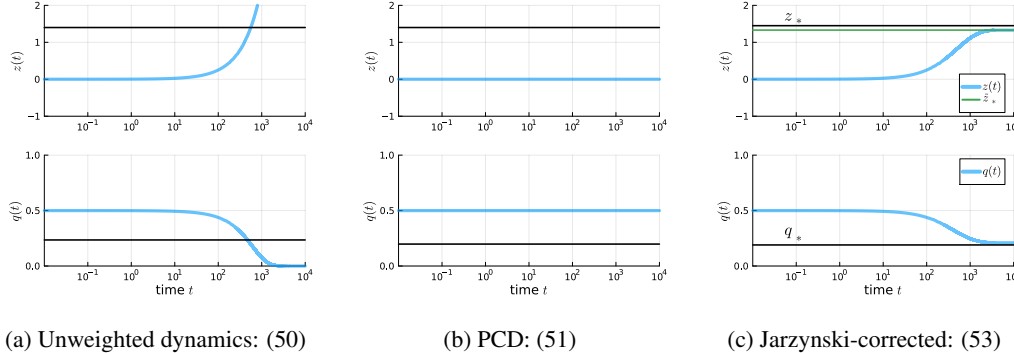

(a) Unweighted dynamics: (50)     (b) PCD: (51)     (c) Jarzynski-corrected: (53)

Figure 14: Behaviors of the three learning algorithms leading to the ODEs (51), (50), and (53) The solid black lines represent the target parameters, $z_*$ on top and $q_* = e^{-z_*}/(1 + e^{-z_*})$ on the bottom. The blue lines show $z(t)$ (top) and the corresponding $q(t) = e^{-z(t)}/(1 + e^{-z(t)})$ (bottom). Figure 14b: PCD algorithm, which leads to no-learning, with the weights not evolving at all in agreement with the behavior predicted by (51). Figure 14a: Dynamics without weights, that leads to mode collapse, with $z(t)$ diverging towards $+\infty$ in agreement with (50) when the parameter $\hat{\gamma} = \hat{q}(0) - \hat{q}_*$ is positive. Figure 14c: Jarzynski-corrected dynamics with weights included, (53). The green line represents (54) and features the small $\log[\hat{q}(0)/\hat{p}(0)]$ correction due to the stochastic fluctuations at initialization between the walkers and the model. In all cases, we used 200 walkers and data points. The continuous-time dynamics were discretized using the Euler-Maruyama scheme with a step-size $h = 0.01$ up to time $T = 10^4$.

- The continuous-version of Contrastive-Divergence is equivalent to the well-known Score-Matching technique ([24]). In this context, (48) is a perturbation of

$$\dot{z}_{\mathrm{cd}}(t) = 0, \tag{52}$$

leading to the same 'no-learning' phenomenon.

- In the model-initialized algorithm using the Jarzynski correction, (48) is a perturbation of

$$\dot{z}_{\mathrm{jar}}(t) = \frac{\hat{q}(0)e^{-z_{\mathrm{jar}}(t)}}{\hat{p}(0) + \hat{q}(0)e^{-z_{\mathrm{jar}}(t)}} - \frac{e^{-\hat{z}_*}}{1 + e^{-\hat{z}_*}} \tag{53}$$

This system has a unique stable point $\tilde{z}_*$ satisfying $e^{-\tilde{z}_*} = \hat{p}(0)e^{-\hat{z}_*}/\hat{q}(0)$, hence

$$\tilde{z}_* = \hat{z}_* + \log\left(\frac{\hat{q}(0)}{\hat{p}(0)}\right). \tag{54}$$

Note that the second term at the right hand side is a small correction of order $O(n^{-1/2})$ since $\hat{q}(0)/\hat{p}(0) = 1 + O(n^{-1/2})$.

These predictions are in good agreement with the results of the simulations shown in Figure 14c.

## C.2 Analysis of (47)

**Using the Jarzynski correction**

In continuous time, the Jarzynski-correction described in Proposition 2 exactly realizes (47). However, even for simple mixtures of Gaussian densities, the expectations in (43) do not have a simple closed-form that would allow for an exact solution. That being said, when $a$ and $b$ are sufficiently well-separated, the system can be seen as a perturbation of a simpler system whose solution can be analyzed.

First, we note that

$$\partial_z U_z(x) = \frac{e^{-z}e^{-\frac{|x-b|^2}{2}}}{e^{-U_z(x)}}. \tag{55}$$

The main idea of the approximations to come is that $\partial_z U_z(x)$ is almost zero when $x$ is far from $b$ (and in particular, close to $a$), and is almost 1 if $x$ is close to $b$. The next lemma quantifies this; from now on we will adopt the notation

$$I_a = [a-4, a+4] \qquad \text{and} \qquad I_b = [b-4, b+4]. \qquad (56)$$

**Lemma 1.** *Under Assumption* (A1), *for any* $v \in \mathbb{R}$,

- *if* $x \in I_a$, *then* $\partial_z U_v(x) \le e^{-v-10}$;

- *if* $x \in I_b$, *then* $|\partial_z U_v(x) - 1| \le e^{v-10}$.

*Proof.* From (55), we see that if $x \in I_a$ then $|x-b| > 6$ and consequently $e^{-(x-b)^2/2} \le e^{-18}$. But the denominator of (55) is itself greater than $e^{-(x-a)^2/2}$ which is itself greater than $e^{-4^2/2} = e^{-8}$ since $|x-a| < 4$. This gives the first bound and the second is proved similarly. $\qquad \square$

We recall that if $\xi \sim \mathcal{N}(0,1)$, then $\mathbb{P}(|\xi| > t) \le e^{-t^2/2}/t$, hence

$$\mathbb{P}(|\xi| > 4) \le e^{-4^2/2}/4 \le 0.0001. \qquad (57)$$

**Lemma 2.** *Let* $u, v \in \mathbb{R}$. *Under Assumption* (A1), *we have*

$$\left| \mathbb{E}_u[\partial_z U_z|_{z=v}] - \frac{e^{-u}}{1+e^{-u}} \right| \le \varepsilon \qquad (58)$$

*where* $|\varepsilon| \le 0.0002 + 2e^{-10}e^{|v|}$.

*Proof.* The integral is exactly given by

$$\frac{1}{1+e^{-u}} \mathbb{E}\left[ \frac{e^{-v-(\xi_a-b)^2/2}}{e^{-U_v(\xi_a)}} \right] + \frac{e^{-u}}{1+e^{-u}} \mathbb{E}\left[ \frac{e^{-v-(\xi_b-b)^2/2}}{e^{-U_v(\xi_b)}} \right] \qquad (59)$$

where $\xi_a, \xi_b$ denote two Gaussian random variables with respective means $a, b$. By (57), $\xi_a$ and $\xi_b$ are respectively contained in $I_a = [a-4, a+4]$ and $I_b = [b-4, b+4]$ with probability greater than 0.999. Let us examine the first term of (59). The fraction inside the expectation is always smaller than 1, hence by Lemma 1,

$$\partial_z U_z(\xi_a)|_{z=v} \le \mathbf{1}_{\xi_a \notin I_a} + \mathbf{1}_{\xi_a \in I_a} \partial_z U_z(\xi_a)|_{z=v} \le \mathbf{1}_{\xi_a \notin I_a} + e^{-v-10}.$$

Consequently, by (57), the expectation is smaller than $0.0001 + e^{-v-10}$. The second expectation in (59) is equal to

$$1 - \mathbb{E}\left[ \frac{e^{-(X_b-a)^2/2}}{e^{-U_v(\xi_b)}} \right]$$

and by the same kind of analysis, the expectation here is smaller than $0.0001 + e^{v-10}$. Gathering the bounds yields the result. $\qquad \square$

In particular, the right hand side of (47) can be approximated by $\frac{e^{-z(t)}}{1+e^{-z(t)}} - \frac{e^{-z_*}}{1+e^{-z_*}}$ up to an error term smaller than $0.0004 + 2e^{-10}(e^{|z(t)|} + e^{|z_*|})$. If $z(t), z_*$ are contained in a small interval $[-C, C]$ with, say, $C < 5$, this error term is uniformly small in time, and one might see (47) as a perturbation of the following system:

$$\dot{z}(t) = \frac{e^{-z(t)}}{1+e^{-z(t)}} - \frac{e^{-z_*}}{1+e^{-z_*}}, \qquad (60)$$

a system with only one fixed point at $z(t) = z_*$, the ground-truth solution.

**Mode collapse in absence of Jarzynski corection**

Now let us analyze in a similar fashion the dynamics without reweighting. Here, (47) is replaced by

$$\dot{z}(t) = \mathbb{E}_{z(t)}[\partial_z U_{z(t)}(X_t)] - \mathbb{E}_*[\partial_z U_{z(t)}], \tag{61}$$

where the process $X_t$ solves

$$dX_t = -\alpha \nabla U_{z(t)}(X_t)dt + \sqrt{2\alpha}dW_t, \qquad X_0 \sim \rho_{z(0)}$$

The probability density function $\rho(t, x)$ of $X_t$ satisfies a Fokker-Planck equation

$$\partial_t \rho = \alpha \nabla \cdot \left( \nabla U_{z(t)}(x)\rho + \nabla\rho \right), \quad \rho(t = 0) = \rho_{z(0)}$$

which, in full generality, is hard to solve exactly, and thus exact expressions for the first term of (61) are intractable. However, depending on whether $X_0$ is close to $a$ or $b$, the process $X_t$ can be well approximated by an Ornstein-Uhlenbeck process, hence $\rho(t, x)$ can itself be approximated by a Gaussian mixture.

**Proposition 3.** *Suppose that* (A1) *holds and that*

$$\exists T, C \in \mathbb{R}_+ \text{ such that for all } t \in [0, \alpha^{-1}T], \qquad z(t) \in [-C, C]. \tag{A2}$$

*Then one has $D_{\mathrm{KL}}(\rho(0)|\rho(t)) \leq \delta t$ where $\delta = 0.000025 + 100e^{-20}e^{2C}$.*

In other words, $\rho(t)$ is approximately constant, up to reasonnable time scales $t = O(1/\delta)$.

*Proof.* $X_0$ is drawn from $\rho_{z(0)}$, a Gaussian mixture; the probability of it being sampled from a Gaussian with mean $a$ is $e^{-z(0)}/(1 + e^{-z(0)}) = 1/2$. We will work conditionally on this event $\mathcal{E}_a$ and we will note $\rho^a(t)$ the density of $X_0$ conditional on $\mathcal{E}^a$; thus, $\rho^a(0) = \mathcal{N}(a, 1)$. We set $V(x) = \frac{1}{2}|x - a|^2$ so that $\nabla V(x) = (x - a)$ and we consider the following Ornstein-Uhlenbeck process:

$$dY_t = -\alpha \nabla V(Y_t)dt + \sqrt{2\alpha}dW_t, \qquad Y_0 = X_0$$

whose density will be denoted $\tilde{\rho}^a(t)$. We use classical bounds on the divergence between $\rho^a(t)$ and $\tilde{\rho}^a(t)$. For example, the bounds in [74, Lemma 2.20] directly apply and yield

$$D_{\mathrm{KL}}(\tilde{\rho}^a(t)|\rho^a(t)) \leq \frac{1}{4} \int_0^t \mathbb{E}\left[ |\nabla U_{z(s)}(Y_s) - \nabla V(Y_s)|^2 \right] ds.$$

Since $Y_t$ is nothing but an Ornstein-Uhlenbeck at equilibrium, $Y_t \sim \mathcal{N}(a, 1)$ for all $t \geq 0$. The term inside the integral is a Gaussian expectation and will be shown to be small:

$$\mathbb{E}\left[ |\nabla U_{z(s)}(Y_s) - \nabla V(Y_s)|^2 \right] \leq 0.0001 + 400e^{-2z(t)-20}. \tag{62}$$

Consequently,

$$D_{\mathrm{KL}}(\tilde{\rho}^a(t)|\rho^a(t)) \leq t\frac{0.0001}{4} + 100e^{-20} \int_0^t e^{-2z(s)}ds.$$

Under (A2), the overall bound remains smaller than $t$ times $0.000025 + 100e^{-20}e^{2C}$ as requested, thus proving that $\tilde{\rho}^a(t) = \mathcal{N}(a, 1)$ and $\rho^a(t)$ are close with the same quantitative bound

Similarly, $\rho^b(t)$, the density of $X_t$ conditional on $X_0$ being sampled from a Gaussian with mean $b$, is close to $\mathcal{N}(b, 1)$ with the same quantitative bounds.

Overall, using the chain rule for KL divergences,

$$D_{\mathrm{KL}}(\tilde{\rho}(t)|\rho(t)) \leq \mathbb{P}(\mathcal{E}_a)D_{\mathrm{KL}}(\tilde{\rho}^a(t)|\rho^a(t)) + \mathbb{P}(\overline{\mathcal{E}_a})D_{\mathrm{KL}}(\tilde{\rho}^b(t)|\rho^b(t)) \leq \delta t.$$

In other words, $\rho(t)$ is close to a mixture of two Gaussians with modes centered at $a, b$, and the probability of belonging to the first mode is the probability of $X_0$ belonging to the first mode, that is, $e^{-z(0)}/(1 + e^{-z(0)}) = 1/2$. □

*Proof of* (62). We have

$$\nabla U_z(x) = \frac{(x - a)e^{-|x-a|^2/2} + (x - b)e^{-|x-b|^2/2-z}}{U_z(x)}. \tag{63}$$

Using Lemma 1 and the fact that if $x \in I_a$ then $|x - b| < 16$ and $|x - a| < 4$, we get $|\nabla U_v(x) - (x - a)| \leq 20\varepsilon$ with $\varepsilon \leq e^{-v-10}$. Consequently,

$$
\begin{aligned}
\mathbb{E}\left[|\nabla U_{z(s)}(Y_s) - \nabla V(Y_s)|^2\right] &\leq \mathbb{P}(Y_t \notin I_a) + (20e^{-v-10})^2 \\
&\leq 0.0001 + 400e^{-2v-20}.
\end{aligned}
$$

$\square$

As a consequence of the Proposition 3, the first term of (61) can be approximated by $\mathbb{E}_{z(0)}[\partial_z U_{z(t)}(X_t)] = \mathbb{E}_{z(0)}[\partial_z U_{z(t)}]$, which in turn can be approximated by $e^{-z(0)}/(1 + e^{-z(0)})$ thanks to (58). Overall, (61) is therefore a perturbation of the system

$$
\dot{z}(t) = \frac{e^{-z(0)}}{1 + e^{-z(0)}} - \frac{e^{-z_*}}{1 + e^{-z_*}} = \frac{1}{2} - q_* =: \gamma.
$$

Since the right hand side no longer depends on $z(t)$, this system leads to a constant drift of $z(t)$, that is $z(t) = \gamma t$, leading to mode collapse since $(1 + e^{-z(t)})^{-1} \approx (1 + e^{\gamma t})^{-1}$ goes to either 0 or 1.

**Mode collapse for Contrastive-Divergence and Score-Matching**

The continuous-time limit of Contrastive Divergence (Algorithm 3) is equivalent to Score-Matching minimization ([28]). The objective function becomes the Stein score,

$$
\begin{aligned}
\mathrm{SM}(z) &= \mathbb{E}_*[|\nabla \log \rho_z(X) - \nabla \log \rho_{z_*}(X)|^2] \\
&= \mathbb{E}_*[|\nabla U_z(X) - \nabla U_{z_*}(X)|^2],
\end{aligned}
$$

which is in theory intractable due to the presence of the unknown parameter $z_*$; a well-known computation from [24] shows that the gradient $\partial_z \mathrm{SM}(z)$ can be estimated using the training samples without resorting to $z_*$. Note that in this context, there are no 'walkers'.

Now, the dynamics (47) is replaced by

$$
\dot{z}(t) = \partial_z \mathbb{E}_*[|\nabla U_{z(t)}(X) - \nabla U_{z_*}(X)|^2] \tag{64}
$$

$$
= p_* \partial_z \mathbb{E}[|\nabla U_{z(t)}(\xi_a) - \nabla U_{z_*}(\xi_a)|^2] + q_* \partial_z \mathbb{E}[|\nabla U_{z(t)}(\xi_b) - \nabla U_{z_*}(\xi_b)|^2] \tag{65}
$$

where here again $\xi_x \sim \mathcal{N}(x, 1)$. From (62) and the triangle inequality, we have

$$
\begin{aligned}
&\left(\mathbb{E}[|\nabla U_{z(t)}(\xi_a) - \nabla U_{z_*}(\xi_a)|^2]\right)^{1/2} \leq \\
&\left(\mathbb{E}[|\nabla U_{z(t)}(\xi_a) - (\xi_a - a)|^2]\right)^{1/2} + \left(\mathbb{E}[|\nabla U_{z_*}(\xi_a) - (\xi_a - a)|^2]\right)^{1/2} \\
&\leq 0.0002 + 800e^{-2z(t)-20}.
\end{aligned}
$$

and the same approximation holds for the second part in (65). Overall, we get that for any reasonnable $z$, $\mathrm{SM}(z) \approx 0$: that is, every $z$ minimizes the score. A similar analysis leads to $\partial_z \mathrm{SM}(z) \approx 0$. Consequently, $\dot{z}(t) \approx 0$: Score Matching and Contrastive Divergence leads to 'no-learning'.

### C.3 Empirical gradient descent analysis of (48)

The gradient descent (47) represented an ideal situation where the expectations $\mathbb{E}_{z(t)}, \mathbb{E}_*$ can be exactly analyzed. In practice, the two terms of (47) are estimated; the second term using a finite number of training data $\{x_*^i\}$, and the first one using a finite number of walkers $\{X_t^i\}$, with associated weights $e^{A_t^i}$ which are either evolved using the Jarzynski rule, or simply set to 1 in the PCD algorithm. Our goal in this section is to explain how these finite-size approximations do not substantially modify the previous analysis and lead to the behaviour presented in C.1. For simplicity we keep the time continuous.

We recall (48):

$$
\dot{z}(t) = \frac{\sum_{i=1}^N e^{A_t^i} \partial_z U_{z(t)}(X_t^i)}{\sum_{i=1}^N e^{A_t^i}} - \frac{\sum_{i=1}^n \partial_z U_{z(t)}(x_*^i)}{n}. \tag{66}
$$

**The dynamics with Jarzynski correction leads to the correct estimation of the empirical weigths**

The continuous-time dynamics of the walkers and weigths in our method is given by

$$dX_t^i = -\alpha \nabla U_{z(t)}(X_t^i)dt + \sqrt{2\alpha}dW_t \tag{67}$$

$$\dot{A}_t^i = -\partial_z U_{z(t)}(X_t^i)\dot{z}(t). \tag{68}$$

Let $\hat{n}_*^a$ be the number of training data in $I_a$ and $\hat{p}_* = \hat{n}_*^a/n$ their proportion, and similarly $\hat{q}_*$ the proportion in $I_b$, and $r = 1 - \hat{p}_* - \hat{q}_*$. By elementary concentration results, the remainder $1 - \hat{p}_* - \hat{q}_*$ can be neglected: with high probability, it is smaller than, eg, 0.0001. We will note $\hat{z}_*$ the parameter satisfying $\hat{q}_* = \frac{e^{-\hat{z}_*}}{1+e^{-\hat{z}_*}}$. Using Lemma 1, the second term in (66) is approximated by $\hat{q}_*$. Now let us turn to the first term in (66). Still using Lemma 1, we see that the first term in (66) is well approximated by

$$\frac{\sum_{i:\, x_t^i \in I_b} e^{A_t^i}}{\sum_{i=1}^n e^{A_t^i}}. \tag{69}$$

The second equation in (67) entails $e^{A_t^i} = \exp\left(-\int_0^t \partial_z U_{z(s)}(X_s^i)\dot{z}(s)ds\right)$. Now let us use Lemma 1: if $X_s^i \in I_a$, then $\partial_z U_{z(s)}(X_s^i) \approx 0$. Conversely, if $X_s^i \in I_b$, then $\partial_z U_{z(s)}(X_s^i) \approx 1$. Moreover, Proposition 3 and its proof essentially show that if $X_0^i$ belongs to the first well (close to $a$), then with high probability so does $X_s^a$ for every $s$, and in particular $X_s^a \in I_a$ with high probability for every $s$. Consequently,

$$e^{A_t^i} \approx \exp\left(-\int_0^t 0 ds\right) = 1 \qquad\qquad\qquad\qquad \text{if } X_0^i \in I_a, \tag{70}$$

$$e^{A_t^i} \approx \exp\left(-\int_0^t \dot{z}(s)ds\right) = \exp\left(-z(t) + z(0)\right) = \exp\left(-z(t)\right) \qquad \text{if } X_0^i \in I_b. \tag{71}$$

As already explained, the dynamics (67) leaves approximately constant the number of walkers in both modes; consequently, the proportion $\hat{q}(t)$ of walkers $X_t^i$ in $I_b$ remains well approximated by the initial proportion, which is $\hat{q}(0)$, and we obtain that (69) is well approximated by

$$\frac{\hat{q}(0)e^{-z(t)}}{\hat{p}(0) + e^{-z(t)}\hat{q}(0)} \tag{72}$$

where we noted $\hat{p}(0) = 1 - \hat{q}(0)$. Note that since $z(0) = 0$, with high probability $\hat{p}(0)$ and $\hat{q}(0)$ are close to $1/2$. The random variable $\hat{p}(0)/\hat{q}(0)$ is thus close to 1.

Overall, we obtain that the system (66) is a perturbation of the following system:

$$\dot{z}(t) = \frac{\hat{q}(0)e^{-\hat{z}(t)}}{\hat{p}(0) + \hat{q}(0)e^{-\hat{z}(t)}} - \frac{e^{-\hat{z}_*}}{1 + e^{-\hat{z}_*}}. \tag{73}$$

This system has a unique stable point equal to

$$\tilde{z}_* := \hat{z}_* + \log(\hat{q}(0)/\hat{p}(0)). \tag{74}$$

*Remark.* If the algorithm had been started at $z(0) \neq 0$, one could check that the stable point would become $\hat{z}_* + \log(\hat{q}(0)/\hat{p}(0)) + z(0)$.

**Freezing the weights leads to mode collapse**

If the walkers still evolve under the Langevin dynamics in (67) but the weigths are frozen at $e^{A_t^i} = e^0 = 1$ ('unweighted' algorithm), then (66) becomes

$$\dot{z}(t) = \frac{\sum_{i=1}^N \partial_z U_{z(t)}(X_t^i)}{N} - \frac{\sum_{i=1}^n \partial_z U_{z(t)}(x_*^i)}{n}. \tag{75}$$

Keeping the same notation as in the last subsection, the second term is still approximated by $\hat{q}_*$, but this time the first term is instead approximated by

$$\frac{\sum_{i:X_t^i \in I_b} 1}{n} = \hat{q}(t) \approx \hat{q}(0).$$

Consequently, (75) is a perturbation of the system

$$\dot{z}(t) = \hat{q}(0) - \frac{e^{-\hat{z}_*}}{1 + e^{-\hat{z}_*}} =: \hat{\gamma}, \tag{76}$$

which no longer depends on $t$ and thus leads to $z(t) = \hat{\gamma}t$ and to mode collapse.

**The PCD algorithm leads to no-learning**

In the preceding paragraph, at initialization, the walkers $X_0^i$ are distributed according to the initial model $\rho_{z(0)}$. In the PCD algorithm 4, the walkers are instead initialized directly at the training data $\{x_*^i\}_{i=1}^n$. However, in this case, the analysis of the preceding paragraph remains essentially the same, with a single difference: the initial proportion of walkers that are close to $a$, noted $\hat{q}(0)$, is now exactly $\hat{q}_*$. Thus, (66) becomes a perturbation of

$$\dot{z}(t) = \hat{q}_* - \hat{q}_* = 0. \tag{77}$$

The parameters remain constantly equal to its initial value $z(0)$, i.e. there is no learning.

**The CD algorithm leads to no-learning**

The Continuous-time Contrastive-Divergence algorithm is minimizing the Stein score and is equivalent to Score-Matching as mentioned above: the direction of the gradient of the log-likelihood is that of the gradient of the Stein Score, leading to no-learing. With the estimation given by the training samples, the analysis is exactly the same as above:

$$\dot{z}(t) = \frac{1}{n} \sum_{i=1}^n \partial_z |\nabla U_z(x_*^i) - \nabla U_{z_*}(x_*^i)|^2$$

$$= \frac{1}{n} \sum_{i=1}^n 2 \partial_z \nabla U_z(x_*^i) \times (\nabla U_z(x_*^i) - \nabla U_{z_*}(x_*^i)).$$

If $x_*^i \in I_a$, then as explained in the proof of Lemma 62, $\nabla U_s(x_*^i) \approx (x_*^i - a)$ for every $s$, hence $\partial_z \nabla U_z(x_*^i) \times (\nabla U_z(x_*^i) - \nabla U_{z_*}(x_*^i)) \approx 0$. The same holds for $x_*^i \in I_b$, leading to (66) being a perturbation of the system

$$\dot{z}(t) = 0. \tag{78}$$

## C.4 On mode-collapse oscillations

Most of the approximations performed earlier rely on (A2), that is, the learned parameter $z(t)$ remains in a compact set.

However, in the unweighted algorithm, this is no longer the case at large time scales, since $z(t)$ diverges from (76); in particular, the approximations from Lemma 1 become meaningless. In fact, Proposition 3 is no longer relevant. The core of Proposition 3 rests upon the fact that if a walker $X_t^i$ is close to $a$, then its dynamics (67) is close to an Ornstein-Uhlenbeck process since $\nabla U_{z(t)}(X_t^i) \approx (X_t^i - a)$. This fails when $z(t)$ has a large absolute value. Let us suppose for instance that $z(t)$ is very small (negative), so that $e^{-z(t)}$ is very large, $|z(t)| \gg |x - b|^2$. In (63), the first term of the numerator is dominated by the second term. Overall, we get

$$\nabla U_{z(t)}(X_t^i) \approx (X_t^i - b),$$

and this is valid for all $X_t^i$. Consequently, *all* the walkers now undergo an Ornstein-Uhlenbeck process *centered at $b$* and in particular, the walkers that are close to $a$ are exponentially fast transferred to the region close to $b$. At this point, the first term in (75) becomes close to 1, leading to the approximated system $\dot{z}(t) = 1 - \hat{q}_*$: $z(t)$ oscillates back to $+\infty$, until the same phenomenon happens again and all the walkers transfer to the region close to $a$.

This leads to an oscillating behavior that can be observed on longer time scales (see Figure 8 for example). We do not think that this phenomenon is relevant to real-world situations since most learning algorithms are trained for a limited time period.

