# OpenReview forum: "Efficient Training of Energy-Based Models Using Jarzynski Equality"
_NeurIPS.cc/2023/Conference — NeurIPS 2023 poster_

### Official Review · Reviewer_nHa5 · 2023-07-02

**Soundness:** 3 good
**Presentation:** 3 good
**Contribution:** 3 good
**Rating:** 4
**Confidence:** 4

**Summary:**

This paper uses Jarzinsky Equality to analyze the training of Energy Based Model (EBM) using cross entropy loss. The authors highlight the problem of directly approximating the gradient using the unweighted mean of nonequilibrium samples and propose a modified algorithm based on reweighting.

**Strengths:**

I like this idea of using Jarzinsky Equality to analyze the current training problem existing in EBM trained with cross entropy loss. I think the analysis is reasonable and hit the core problem of EBM training.

**Weaknesses:**

However, I remain unconvinced by the proposed algorithm, particularly regarding its potential for practical scalability. I have the following concerns:

1. Sample inefficiency: The proposed method utilizes a reweighted version of gradient updates. While this approach may be theoretically sound for calculating expectations, I am apprehensive about its sample efficiency. Specifically, the reweighting term takes the form of an exponential, which can rapidly become either very large or very small. Consequently, a large batch of samples may need to be maintained and updated at every step. Additionally, with more complex datasets, an increased number of individual chains/walkers may be required, posing scalability challenges for the model.

2. Moreover, the implementation necessitates the tracking of additional information such as $A_{t}^{i}$ and $X_{t}^{i}$. This could impose significant computational and memory burdens.

3. Insufficient experiments: The only complex dataset showcased in this work is a subset of MNIST, containing only three digits. Furthermore, no qualitative measures of overall sample quality, such as FID or Inception Score, are presented. Consequently, with the current results, it is difficult for me to believe that the proposed methods can outperform the original CE and yield satisfactory results. It will be more convincing if the authors can justify their algorithm on more complex dataset like cifar10 or cifar100.

**Questions:**

1. For the resampling step, how to assign new value to each walker $X_{kr}^{i}$ that needs to be resampled? Do you just resample $X_{kr}^{i}$ from $\rho_{\theta0}$? Also why can you reset $A{kr}^{i}=0$? For my understanding, according to eq5 and eq14, you should keep track of the whole chain of $A_t$ and $X_t$ through the training and sampling process, or the results might be incorrect/

2. How can you generate new samples after the model is trained? I ask this because there can be a propblem that an EBM model trined with persistent MCMC have difficulty in generating new samples after the model is trained.

3. The review of related work on EBM training seems to be a bit rough and misses many recent works for EBM training and scaling up. For example:
1. A theory of generative convnet.
2. Flow contrastive estimation of energy-based model.
3. Your classifier is secretly an energy based model and you should treat it like one
4. Vaebm: A symbiosis between variational autoencoders and energy-based model.
5. Generalized energy based mode.
6. A tale of two flows: Cooperative learning of langevin flow and normalizing flow toward energy-based model.
7. Guiding energy-based models via contrastive latent variable.

**Limitations:**

The authors have adequately addressed the limitations.

---

> ### Author Rebuttal · Authors · 2023-08-09
>
> **Sampling efficiency.** As with our method, the sample size (i.e. the number of wakers) must be large enough to guarantee statistical accuracy. There are few theoretical guarantees about this issue, unfortunately, and it is known that the Jarzynski weight can indeed become large if the sample size is too small, which impacts the accuracy. We stress however that this problem can be mitigated because these weights allow us to estimate the effective sample size (ESS) (which other methods cannot easily do), and the resampling step can be used to control these weights along with the ESS.
>
> We also stress that our method also allows for mini-batching (see Algorithm 4 in the paper), as explained in our reply to Reviewer d8UC. As stated in this reply, the mini-batching procedure  is very simple:
> 1. Store the position and the scalar weights of all the walkers;
> 2. At each step of SGD, evolve only the position and the weights of a subset of walkers, while keeping the others untouched in a replay buffer;
> 3. When a walker is picked from the replay buffer, first adjust its weight to the current value of $U_\theta$ (which only requires calculating this energy and not its gradient), then evolve its position and weight.
>
> This makes the cost of our method comparable to that of the CD or PCD algorithm, though the mini-batching is better justified in our method than in PCD: Indeed, not evolving all the walkers in the latter increases the bias (that is, the mismatch between the empirical distribution of these walkers and the Gibbs distribution associated with the current $U_\theta$); in contrast, our approach removes this bias using the weights *whether the walkers evolve (i.e. are part of the mini-batches) or not*.
>
> **Variables to track.** Tracking the position $X^i_k$ of the walkers in our method is not different than tracking their positions in the CD and the PCD algorithms. Compared to these methods, we must additionally track the values of the weights $A^i_k$: we stress however that these weights are *scalar* quantities, so there is no noticeable extra cost in storing them. There is also no extra cost from evolving the weights in the mini-batch since this operation uses the gradient $\nabla U_\theta(X_k)$ which is already needed to evolve the positions of walkers in the mini-batch (note that this gradient is also needed in the CD and the PCD algorithms).
>
> **Insufficient experiments.** As stated in the common reply, we did not have the  computing resources necessary to perform experiments on large datasets such as CIFAR-10 or ImageNet, unfortunately. However, we plan to perform these experiments in collaboration with other groups with more computational resources in a follow-up paper.
>
> As also stated in the Common Reply, we did perform additional standard tests on the experiments we have, and now report FID for the MNIST experiments along with ablation studies involving varying the number of walkers and the size of the mini-batches, see Fig. 2 in the Common Reply (this is also done in the GMM experiments, see Fig. 1). In addition, we performed new experiments on a high-dimensional discrete data set, see the description in the Common Reply and Fig. 3.
>
> **Questions:**
> * *Resampling step.* The resampling step involves splitting walkers with large weights, and killing those with low weights. This operation is done within the current population of the walkers (i.e. it does not involve resampling from an external distribution such as $\rho_{\theta_0}$ or from the data set, but rather from the empirical distribution of the walkers). This step is performed to control the ESS, and it is standard in sequential Monte Carlo methods: after resampling, all the values of all the log-weights $A^i_k$ are reset to zero, i.e. the ESS is again the total number of walkers (though other strategies evolving partial resampling are also possible). We also stress that the resampling step only involves selecting walkers using their weights. Since these weights are scalar quantities, the cost of this step is independent of the dimensionality of the problem.
>
> * *Generation of new samples.* In contrast with other methods (like the CD or PCD algorithm, or their variant involving persistent MCMC), at all iterations during training the weighted walkers in our approach sample the EBM by construction. As a result, they can directly be used as new samples, and we can keep evolving them after the training stops to get more new samples. In the procedure, these samples are weighted, but all weights become identical after resampling: these "resampled samples" are the ones we can use e.g. as generated images (as we did in the MNIST experiments, see Fig. 2 in the paper). In this sense, our approach truly is an implementation of an EBM that can be used as a generative model.
>
> * *Review of the literature.* We thank the reviewer for pointing out these references to us: we will cite them in the new version of our paper. As far as we can tell, these papers propose variants of EBM training that combine it with other techniques, but they do not guarantee that these methods perform training by effectively minimizing the KL divergence. We believe that our method could be used in concert with some of these approaches to remedy this deficiency.

---

> > ### Comment · Reviewer_nHa5 · 2023-08-14
> >
> > While I do appreciate the replies given by the authors, I have to say I'm not fully convinced by it. My primary concern continues to be the scalability of the proposed method. As outlined in my earlier review, I can see some shortcomings in the method, with sample efficiency being the most pronounced. While the authors suggest potential solutions such as the approximation in Algorithm 4, the practical implications of such approximations on training remains uncertain to me. A compelling way to assuage these concerns would be to present experimental results on more intricate datasets. (As also mentioned by other reviewers, the MNIST dataset is too simple.)
> >
> > I recognize that developing a state-of-the-art model can be resource-intensive. However, based on my experience, achieving functional results (or at least generating valid samples comparable with foundational baselines) on $32 \times 32$ resolution images like CIFAR-10 shouldn't pose a significant challenge. For instance, [1] offers a one page code capable of training on a single GPU, possibly producing valid samples within a day. For me, the absence of such results raises further doubts about the efficiency and scalability of the proposed method.
> >
> > In summary, while I appreciate the core concept of this paper, the present findings leave me hesitant to affirm its efficacy. Given these reservations, I'm inclined to maintain my initial rating.
> >
> >
> > [1] On learning non-convergent non-persistent short-run MCMC toward energy-based model.

---

> > > ### Author Response · Authors · 2023-08-20
> > > **New experiments on the CIFAR-10 image dataset**
> > >
> > > **CIFAR-10 Experiments**
> > > Following the Reviewers suggestion, we performed an empirical evaluation of our method on the full CIFAR-10 (32 $\times$ 32) image dataset. We used the setup of Ref. [1], with the same neural architecture with $n_f = 64$ features, and compared the results obtained with our approach with mini-batching (Algorithm 4) to those obtained with PCD and PCD with data augumentation of Ref. [2] (which consists of a combination of color distortion, horizontal flip, rescaling, and Gaussian blur augumentations, to help the mixing of the MCMC sampling and stabilize training).
> > >
> > > The code used to performed these new experiments is available (along with some generated images) in the anonymized github referenced in our paper. The hyperparameters are the same in all cases: we take $N = 4096$ Langevin walkers with a mini-batch size $N' = 256$. We use the Adam optimizer with learning rate $\eta = 10^{-4}$ and inject a Gaussian noise of standard deviation $\sigma = 3\times 10^{-2}$ to the dataset while performing gradient clipping in Langevin sampling for better performance. All the experiments were performed on a single A100 GPU. Training for 600 epochs took about 22 hours  with the PCD algorithm (w/ and w/o data augumentation) and about 23 hours with our method.
> > >
> > >
> > > The results in terms of FID and IS are:
> > > | Method     | FID   | IS
> > > | -------    |------ |------
> > > |PCD |45.63 | 5.62
> > > |PCD with data augmentation |42.27| 6.34
> > > |Our method (Algorithm 4)| 38.49 | 6.76
> > >
> > > We will add these new results to the revised version of our paper. They indicate that our method can achieve slightly better performance than the PCD algorithms w/ and w/o data augumentation at a similar computational cost. Furthermore, these results on CIFAR-10 suggest that our method scales well to complicated training tasks on more realistic data sets.
> > >
> > > ***References:***
> > >
> > > [1] Nijkamp, E., Hill, M., Zhu, S. C., & Wu, Y. N. (2019). Learning non-convergent non-persistent short-run MCMC toward energy-based model. Advances in Neural Information Processing Systems, 32.
> > >
> > > [2] Du, Y., Li, S., Tenenbaum, J., & Mordatch, I. (2020). Improved contrastive divergence training of energy-based models. arXiv preprint arXiv:2012.01316.

---

### Official Review · Reviewer_d8UC · 2023-07-05

**Soundness:** 3 good
**Presentation:** 3 good
**Contribution:** 3 good
**Rating:** 7
**Confidence:** 4

**Summary:**

The paper under consideration proposes a modification of the standard EBM training procedure inspired by Jarzynski equality from thermodynamics. The idea is as follows. It is known that EBM training requires sampling from the distribution for which we know only its energy function given by, e.g., neural network $E_{\theta}$. Typically, this is done by ULA algorithm or its analogues. Given current NN parameters $\theta$, we try to sample from $p_{\theta} \sim \exp( - E_{\theta})$ using the corresponding MCMC algorithm and only after that we update the parameters $\theta$. The authors take a step further and propose to change the paradigm. They consider a specific discrete procedure which resembles Annealed Importance Sampling MCMC algorithm. This procedure evolves both particles and model (NN) parameters so that the parameters turn out to follow gradient flow with respect to KL divergence. The authors validate and show the advantages of ther idea on moderate-dimensional Gaussian Mixture setup and MNIST dataset.

**EDIT 2:** During the discussion, the authors came up with additional CIFAR10 experiment, which validate the applicability of their approach. On the one hand, the provided results are far from being SOTA (for example, [1] has much better metrics). On the other hand, the authors utilize the simplest CNNs and, from my point of view, such experiments are already valuable because they show that the method works without tricky architectural and (I guess) hyperparameter engineering. So, it is a good proof-of-concept setup. Based on this, I consider rising my score by one (to **7**) and recommend accepting the paper. At the same time, I would suggest authors to devote some time improving the architectures and tuning hyperparameters in order to achieve the performance close to [1] or better. For now, the paper is more methodological and theoretically-biased. From my point of view, a good NeurIPS paper should not underestimate practical aspects.

[1] Du et. al., Improved Contrastive Divergence Training of Energy-Based Models, http://proceedings.mlr.press/v139/du21b/du21b.pdf

**Strengths:**

The paper seems to present a novel and theoretically sound idea. To be more precise, I would like to stress out the following points:
- In spite of the Annealed Importance Sampling (AIS) is known for the decades, and it is also inspired by Jarzynski equality, the paper under consideration is not just utilizes AIS for sampling from an unnormalized distribution, but proposes a theoretically-grounded and assumptions-free procedure which prescribes how to progress particles along with model parameters to consistently converge to the distribution of interest.
- The proposed method eliminates the problem of slow mixing faced by conventional EBMs methods. It seems that, given large enough particles (walkers), we can just launch the proposed algorithm, and it will succeed in recovering the desired energy function. We don’t need to think about how much steps of Langevin samples should be done for each parameter's update step.
- The authors demonstrate several more-or-less convincing setups that demonstrate the superiority of their method compared to standard CD and PSD approaches. Additionally, for simple 1D setup they provide the theoretical arguments which supports their thesis regarding this supremacy (to be honest, I didn’t get into details of the analysis in Appendix C, but it seems correct).

**Weaknesses:**

From my point of view, the main weaknesses of the paper under consideration are as follows:
- As I understand, the Algorithm 1 can not be fairly adapted for mini-batch setup and requires evolving simultaneously numerous particles (of training dataset order). In other words, for each gradient step w.r.t. model parameters, we need to evolve the whole dataset of points (Algorithm 4 also requires updates for each point from the dataset). From my point of view, it is the really great drawback which limits method applicability.
- Related to the previous point, it is not clear if the proposed approach scales well to more sophisticated setups like CIFAR-10, Celeba etc. Will all these technical resampling procedures (described in Section 3.3)  work well for these tasks? So, at the current stage, I treat the paper as having limited practical validation in terms of plausibly looking images.
- There are no quantitative comparison between proposed method and CD (PCD) for MNIST experiment. Showing just images is not very convincing. Perhaps, the better classes balance is achieved by cost of quality degradation.

**Questions:**

- Line 163: Are there theoretical guarantees that the integral (which defines $\hat{\rho}(t, x)$) converges?
- A natural question from me as a practitioner of EBMs: how many walkers $N$  we need to support in order to Algorithm 1 (or Algorithm 4) stable converge? Some bounds or at least kind of ablation study for your considered problems (MoG and MNIST) would be quite interesting.
- Which parameters of Algorithm 1 were used in the MNIST experiment?
- What is the Assumption (A1) you use in Lemma 1 and Lemma 2 in the appendix? I didn’t manage to find such an assumption in the paper.

Misprints
- Line 170: In the second equation of the inline formula, one should add the integration w.r.t. $x$.
- Line 180: “that that” -> “that”.
- Algorithm 1, line 2: it seems $A_0^i$ should be initialized as zeros.
- Line 433: “given and” -> “given”.
- Formula (51), denominator: $U_z(x)$ -> $\exp(- U_z(x))$.
- Figure 10 (about MNIST dataset) label refers to formula (25) (which is about MoG experiment). There should be another formula.

**Limitations:**

The potential negative societal impact of the work are not addressed. However, it seems to be the save as for thousands of papers which deals with generative modelling. So I have no special concerns.

---

> ### Author Rebuttal · Authors · 2023-08-09
>
> We would like to thank the reviewer for their careful reading, positive view on the manuscript, and constructive suggestions. Below, we address the main points raised.
>
> **Using mini-batches.** *Algorithm 1 can be easily generalized to use mini-batches*, see Algorithm 4 in the paper, and it does not require evolving all the walkers at every SGD step. The procedure is very simple:
> 1. Store the position and the scalar weights of all the walkers;
> 2. At each step of SGD, evolve only the position and the weights of a subset of walkers, while keeping the others untouched in a replay buffer;
> 3. When a walker is picked from the replay buffer, first adjust its weight to the current value of $U_\theta$ (which only requires calculating this energy and not its gradient), then evolve its position and weight.
>
> Let us also note that using mini-batching is better justified in our method than in the PCD algorithm. Indeed, not evolving all the walkers in the latter increases the bias (that is, the mismatch between the empirical distribution of these walkers and the Gibbs distribution associated with the current $U_\theta$); in contrast, our approach removes this bias using the weights *whether the walkers evolve (i.e. are part of the mini-batches) or not*. Of course, the reweighting cannot do miracles if the number of walkers is not large enough -- all sampling methods have the same issue. However, our approach has the advantage that its sample efficiency can be estimated via the calculation of the effective sample size (which involves the weights) and controlled via resampling.
>
> **Scalability of the resampling procedure.** The resampling procedure only involves selecting walkers using their weights. Since these weights are scalar quantities, the cost of this step is independent of the dimensionality of the problem.
>
> **Specific questions**
> * *Line 163: Are there theoretical guarantees that the integral (which defines $\hat\rho_\theta(t,x)$) converges?*
> Since $\int_0^\infty \int_{\mathbb{R}^d} f(t,a,x) dx da = Z_{\theta(t)}/Z_{\theta_0} $ (Eq. 12), this integral can only diverge if $Z_{\theta(t)}$ does. This cannot happen under our assumption that the target density is of the form $e^{-U_*(x)}/Z_*$ with $Z_*<\infty$ if $\theta(t)$ evolves by GD on the KL divergence. We will add a remark to this effect.
> * *Number of walkers needed*  Our method is an instance of a sequential Monte Carlo algorithm: these algorithms are widely used in practice, but they offer few theoretical guarantees, unfortunately. We have however performed more ablation studies on the GMM and MNIST experiments.
> * *Parameters of Algorithm 1 used in the MNIST experiment*
> We use $n = 9800$, $N = 1024$, $h = \texttt{5e-3}$. As for the total number of training steps, we run $800$ epochs with batch size $128$.
> * *What is the Assumption (A1) you use in Lemma 1 and Lemma 2 in the appendix?* Assumption (A1) is defined above line 456. It is defined inline but we will highlight it in a different way to make it clearer.
>
> **Misprints.**
>      Thank you for pointing these out! We will correct them. And, yes, for Fig. 10, it should be equation (21).

---

> > ### Comment · Reviewer_d8UC · 2023-08-11
> > **Answer**
> >
> > I thank the authors for their answers and clarifications. Indeed, Algorithm 4 does not require updating weights for the whole dataset at each minibatch step. At the same time, I still think that for granting better score, more convincing (large-scale) experiments which demonstrates the generative ability of the proposed method are required. So I consider leaving my score unchanged.

---

> > > ### Author Response · Authors · 2023-08-16
> > >
> > > We thank the reviewer for their reply and constructive suggestion. We are now in the process of testing our approach on CIFAR10 using the setup of https://arxiv.org/abs/1904.09770, as suggested by the reviewer, and we hope that these results will be available very soon, in which case we will include them in the paper. At the same time, **we believe that the theoretical results we derive, and the numerical  experiments we already have, already make  a case for our approach that is strong enough to justify publication in NeurIPS.** In particular, to the best of our knowledge our approach is the only one that offers guarantees to train EBM via SGD on the KL divergence, and it does so at not significant extra cost compared to the state-of-the-art methods currently used to train EBM. As a result our approach could be used in concert with many of those methods (which use e.g. replay buffers, add Gaussian noise to data, or perform data augmentation) simply by adding weights to the MCMC walkers.

---

> > > > ### Author Response · Authors · 2023-08-20
> > > > **New experiments on the CIFAR-10 image dataset**
> > > >
> > > > **CIFAR-10 Experiments**
> > > > Following the Reviewers suggestion, we performed an empirical evaluation of our method on the full CIFAR-10 (32 $\times$ 32) image dataset. We used the setup of Ref. [1], with the same neural architecture with $n_f = 64$ features, and compared the results obtained with our approach with mini-batching (Algorithm 4) to those obtained with PCD and PCD with data augumentation of Ref. [2] (which consists of a combination of color distortion, horizontal flip, rescaling, and Gaussian blur augumentations, to help the mixing of the MCMC sampling and stabilize training).
> > > >
> > > > The code used to performed these new experiments is available (along with some generated images) in the anonymized github referenced in our paper. The hyperparameters are the same in all cases: we take $N = 4096$ Langevin walkers with a mini-batch size $N' = 256$. We use the Adam optimizer with learning rate $\eta = 10^{-4}$ and inject a Gaussian noise of standard deviation $\sigma = 3\times 10^{-2}$ to the dataset while performing gradient clipping in Langevin sampling for better performance. All the experiments were performed on a single A100 GPU. Training for 600 epochs took about 22 hours  with the PCD algorithm (w/ and w/o data augumentation) and about 23 hours with our method.
> > > >
> > > >
> > > > The results in terms of FID and IS are:
> > > > | Method     | FID   | IS
> > > > | -------    |------ |------
> > > > |PCD |45.63 | 5.62
> > > > |PCD with data augmentation |42.27| 6.34
> > > > |Our method (Algorithm 4)| 38.49 | 6.76
> > > >
> > > > We will add these new results to the revised version of our paper. They indicate that our method can achieve slightly better performance than the PCD algorithms w/ and w/o data augumentation at a similar computational cost. Furthermore, these results on CIFAR-10 suggest that our method scales well to complicated training tasks on more realistic data sets.
> > > >
> > > > ***References:***
> > > >
> > > > [1] Nijkamp, E., Hill, M., Zhu, S. C., & Wu, Y. N. (2019). Learning non-convergent non-persistent short-run MCMC toward energy-based model. Advances in Neural Information Processing Systems, 32.
> > > >
> > > > [2] Du, Y., Li, S., Tenenbaum, J., & Mordatch, I. (2020). Improved contrastive divergence training of energy-based models. arXiv preprint arXiv:2012.01316.

---

### Official Review · Reviewer_YzoY · 2023-07-06

**Soundness:** 3 good
**Presentation:** 3 good
**Contribution:** 3 good
**Rating:** 7
**Confidence:** 2

**Summary:**

The authors consider EBMs family of algorithms optimized with a cross-entropy objective. They proposed a modification of the unadjusted Langevin algorithm (ULA) to remove bias from the optimization procedure, basically introducing correction coefficients for UCL gradient estimation. Their method is based on Jarzynski equality and is theoretically justified.  Moreover, the paper confirms empirically the method is less biased as it can recover all "modes" and proportions between them from data correctly in contrast to other baselines optimized with the cross-entropy loss.

**Strengths:**

1) The method is theoretically justified and novel to the best of my knowledge.
2) The method can be used to find a normalizing constant and estimate density.
3) The author demonstrates that the method covers "modes" correctly on two toy examples:  a Gaussian distribution with two modes, and a label-unbalanced MNIST dataset. The proposed method successfully recovers all the modes and their respective proportions, which is in contrast with the baseline ULA method. The toy experiment designs are great.
4) The method is easy to implement and extra (compared to ULA) computational cost is low.
5) The paper is well-written and easy to follow.


**Weaknesses:**

1) The empirical evaluation is weak. The paper only considers two toy datasets. It would be great to explore the proposed method on more challenging datasets. However, the authors noticed it themself in the conclusion.
2) The paper did not use standard metrics to measure performance. Along with more complex datasets, it would be great to measure and compare performance with standard metrics such as FID,  Inception score, recall, and precision. I am especially interested in the recall and IS scores as they are most related to how the authors measure correct proportion and mode-covering in their toy experiments.
3) It is not clear how the new method compares in terms of sample quality to baselines (there are no metrics).


**Questions:**

1) The method can be used for density estimation. Is the proposed method beneficial compared to other density estimation methods? Is it worth evaluating it against other methods on the density estimation task?
2) I am interested in a side-by-side plot with true density and density approximation fitted with a neural network. One can use the toy paper example with two Gaussian modes or more complex distributions. Does this experiment make sense?


**Limitations:**

--

---

> ### Author Rebuttal · Authors · 2023-08-09
>
> We would like to thank the reviewer for their careful reading, positive view on the manuscript, and constructive suggestions. Below, we address the main points raised.
>
> **Scope of the numerical experiments.** As stated in the common reply, we did not have the computing resources necessary to perform experiments on large datasets such as CIFAR-10 or ImageNet, unfortunately. However, we plan to perform these experiments in collaboration with other groups with more computational resources in a follow-up paper.
>
> **Usage of standard metrics.** In the new version, we will report the FID score as the metric for evaluating the performance of our model on MNIST, see Figure 2 of our Common reply and see also our reply to reviewer 1pft on why we do not think FID score is not good for measuring the performance of EBM training methods on MNIST. As for the Inception Score, we have also tried that but it did not produce reasonable results because the pre-trained Inception V3 network used for computing the IS is trained on natural images, which are fundamentally different than MNIST images.
> This comparison can be also performed in the GMM experiment, as well as in the new experiments.
>
>
> **Questions about density estimation and direct visualization.** It is not clear how to perform this comparison in general. We did however perform new experiments on a high-dimensional discrete-data set that allows for direct visualization of the learned energies (i.e. the density in logscale) in 2D, see the description and Fig. 3 in the Common Reply.

---

### Official Review · Reviewer_2q1e · 2023-07-07

**Soundness:** 3 good
**Presentation:** 3 good
**Contribution:** 3 good
**Rating:** 6
**Confidence:** 3

**Summary:**

The paper reports an enhcements in the eficiency of the training of EBMs using Jarzinsky equality.

**Strengths:**

The paper has the potential to help understand the training of EBM better through a thermodynamics based approach. It is well written and elaborated.

**Weaknesses:**

The paper shows a more efficient training of EBM. However, it does not explain why such efficient is obtained by performing the modification using the JE. I think the authors should dedicate a section to elaborate from thermodynamics perspective that why such improvements is achieved.

**Questions:**

Can you please explain what would be the thermodynamics systems, States, temperature, thermal reservoir in the context of the training of a EBM?
In a thermodynamics context, the free energy is connected to the work done on the system by means of an inequality. (Delta_F<W). The equality is when the system is slowly varied from a state to another state. In irreversible thermodynamics, according to JE, no matter how fast the systems is transferred from one state to another, the equality remains valid. In JE, the temperature of the systems in equilibrium, T, has a physical meaning and that is the temperature of the heat reservoir with which the system had been in contact. What would be the interpretation of the T in the training steps of the energy based models using JE?
In JE, an average is taken over all possible realizations that drives the thermodynamics system from state A to state B. can you please explain how this averaging is performed during the training of the EBM?
Can you please explain in the context of the thermodynamics, for example using free energy and external work, on why such efficiency is obtained?



**Limitations:**

I think a more physical understanding of why the performed modification enhances the training will have be great for gaining an understanding of the physics of the training.

---

> ### Author Rebuttal · Authors · 2023-08-09
>
> We would like to thank the reviewer for their careful reading, and their suggestion to better explain underpinning from the statistical physics of our approach based on the Jarzynski equality (JE).
>
> **Thermodynamic interpretation.** This interpretation is actually the same as for physical systems: since the EBM energy $E_\theta$ evolves during training, work is being performed on the walkers and we need to account for it if we want to sample the associated Gibbs distribution consistently. This is precisely what our method achieves by associating a scalar weight to every walker that is used to correct the bias introduced by the work from evolving the energy. The gain in efficiency of our method comes from this consistency: in contrast, when one uses e.g. the PCD algorithm or persistent MCMC, we would have to evolve the parameters $\theta$ infinitely slowly (i.e. adiabatically) compared to the walkers to guarantee that the latter exactly samples the distribution associated with the former (which is akin to using thermodynamic integration in the physical systems). Since an infinite time-scale separation cannot be enforced in practice, work is effectively being performed on the system when one uses the PCD algorithm or persistent MCMC, and neglecting the weights results in a bias.  Our approach based on the Jarzynski equality allows one to remove this bias.
>
> We also note that in the setup of the paper (as in usually done with EBM), the temperature of the system is absorbed in the energy and therefore set to one. As for the average over the realizations, they are done in practice by averaging over the number of walkers used in the procedure, possibly using mini batches: this is akin to using a parallel implementation of JE. Our approach can also be generalized by making multiple passes on successive time-lags, which would make it more similar to what is usually done to apply JE, but we found that proceeding as we do with walkers was sufficient for our needs.
>
> For the reader's convenience, we will follow the reviewer's suggestion and add a section in the Appendix discussing this physical interpretation in more detail.

---

> > ### Comment · Reviewer_2q1e · 2023-08-19
> > **Thanks to the Authors for addressing my comments. I keep my score.**
> >
> > Thanks to the Authors for addressing my comments and questions. I keep my score.

---

### Official Review · Reviewer_ripa · 2023-07-20

**Soundness:** 2 fair
**Presentation:** 3 good
**Contribution:** 2 fair
**Rating:** 6
**Confidence:** 5

**Summary:**

One of the main challenges in the training of Energy-Based Models (EBMs) is the design of a tractable objective that allows for low-variance gradient estimation. Indeed, the maximum likelihood objective requires sampling from the current density model, which usually is computationally expensive and biased due to poor mixing properties. Many alternatives have been proposed to alleviate the issues of maximum likelihood objective: contrastive divergence, persistent contrastive divergence, non-convergent MCMC training, etc.

The current paper approaches the optimization problem of the maximum likelihood objective by proposing a new sampling procedure. Once again, the main challenge is to estimate the term

$$\mathbb{E}_{x\sim q_\theta}\partial_\theta U_\theta(x),$$

where $q_\theta(x) \propto \exp(-U_\theta(x))$ is the current density model. After each parameter update, instead of relying on the MCMC sampling from the density, the authors propose to update the current samples using unadjusted Langevin Dynamics and reweight the generated samples to get an estimator of $\mathbb{E}_{x\sim q_\theta}\partial_\theta U_\theta(x)$. Thus, the main development of the paper is Algorithm 1 and Proposition 2 demonstrating that the proposed estimator is consistent (converges to the truth value for the infinite number of samples).

The authors demonstrate the benefits of the proposed algorithm empirically on two target distributions: the mixture of 2 Gaussians, and 3 (out of 10) MNIST digits. For both experiments, the authors demonstrate that their algorithm allows for a better estimate of the relative mass of the modes.

**Strengths:**

- The proposed algorithm and theoretical developments that justify the algorithm are novel and are definitely of interest to the generative modeling community.
- The paper is clearly written, and all the results are well-presented.

**EDIT:** During the discussion, the authors provided experimental results for CIFAR-10. However, I still have the following concerns:
- The baseline in the empirical study for CIFAR-10 demonstrates much worse performance than in the prior work [2].
- The statements I mentioned in my review are still missing references or justifications, e.g., "Among these methods, EBMs are perhaps the simplest and most interpretable" has to be either justified by references (since it's not properly studied in the paper) or removed.
- The biasedness of the estimator has to be mentioned explicitly next to its introduction since it's an important property (downside) for optimization.

Despite these concerns, I recommend the paper for acceptance.

[2] Du, Y., Li, S., Tenenbaum, J., & Mordatch, I. (2020). Improved contrastive divergence training of energy-based models. arXiv preprint arXiv:2012.01316.

**Weaknesses:**

- The empirical evaluation of the proposed algorithm is insufficient to demonstrate the practical benefits of the proposed algorithm. Indeed, the considered experiments are rather toy examples compared to the relevant literature [1,2,3]. Since Algorithm 1 is the main development of the paper, this is a major flaw of the current submission.
- The role of continuous-time analysis is not clear. Namely, Proposition 1 is never used after the introduction in Section 3.1. Thus, it has no immediate practical or theoretical implications. Even though I appreciate the result, I think the paper will benefit from moving this discussion to the Appendix.
- The following statements in the introduction require references to the corresponding works with justifications.
    - “Among these methods, EBMs are perhaps the simplest and most interpretable”. It’s also worth mentioning which properties of EBMs make them the most interpretable models.
    - “methods in this class are provably unable to learn the relative weights of modes when they are separated by low-density regions”
    - “Unlike the Fisher divergence, the cross-entropy is sensitive to the relative weights in multimodal distributions, but it unfortunately leads to training algorithms that are less justified theoretically and more delicate to use in practice”
    - “Unfortunately, the approximations made in these algorithms are uncontrolled and they are known to induce biases similar to those observed with score-based methods”. Here it is not clear which biases of the score-based methods are mentioned.
    - “Score-matching techniques and variants originate from [9, 21, 22]; their shortcoming in the context of EBM training is investigated in [23] and their blindness to the presence of multiple, imbalanced modes in the target density has been known for long and we refer to [24, 25]) for discussions.” In this context, it is necessary to mention [4] (and numerous papers based on it), which is a score-based technique that led to a number of state-of-the-art methods in generative modeling, at the same time, sharing many common properties with EBMs (e.g. density evaluation).

[1] Nijkamp, E., Hill, M., Zhu, S. C., & Wu, Y. N. (2019). Learning non-convergent non-persistent short-run MCMC toward energy-based model. *Advances in Neural Information Processing Systems*, *32*.
[2] Du, Y., & Mordatch, I. (2019). Implicit generation and modeling with energy-based models. *Advances in Neural Information Processing Systems*, *32*.
[3] Du, Y., Li, S., Tenenbaum, J., & Mordatch, I. (2020). Improved contrastive divergence training of energy-based models. *arXiv preprint arXiv:2012.01316*.
[4] Song, Y., Sohl-Dickstein, J., Kingma, D. P., Kumar, A., Ermon, S., & Poole, B. (2020). Score-based generative modeling through stochastic differential equations. *arXiv preprint arXiv:2011.13456*.



**EDIT:** During the discussion period, the authors provided experimental results for CIFAR-10. However, I still have the following concerns:
- The baseline in the empirical study for CIFAR-10 demonstrates much worse performance than in the prior work [3].
- The statements I mentioned in my review are still missing references or justifications, e.g., "Among these methods, EBMs are perhaps the simplest and most interpretable" has to be either justified by references (since it's not properly studied in the paper) or removed.
- The biasedness of the estimator has to be mentioned explicitly next to its introduction since it's an important property (downside) for optimization.

Despite these concerns, I recommend the paper for acceptance.

[3] Du, Y., Li, S., Tenenbaum, J., & Mordatch, I. (2020). Improved contrastive divergence training of energy-based models. arXiv preprint arXiv:2012.01316.

**Questions:**

I have no questions for the authors.

**Limitations:**

Besides poor empirical evaluation, which does not allow for defining the limits of the proposed method, two following points should be discussed.

- The choice of batch size for experiments is worrisome. According to the authors, they used a batch size of $10^5$ for the mixture of two Gaussians and $1024$ for the reduced-MNIST experiment. At the same time [2] uses $128$ negative and positive examples at every iteration.
- The derived estimate is biased. It is a minor point but it is never mentioned in the paper.

---

> ### Author Rebuttal · Authors · 2023-08-09
>
> We would like to thank the reviewer for their careful reading, positive view on the manuscript, and constructive suggestions. Below, we address the main points raised.
>
> **Scope of the experimental results.** Unfortunately we did not have the computing resources necessary to perform experiments on large datasets such as CIFAR-10 or ImageNet. However, we plan to perform these experiments in collaboration with other groups with more computational resources in a follow-up paper.
>
> We also stress that our experiments on MNIST are nontrivial: indeed, by pruning the data set to 3 digits, we make the overlap between the classes smaller and the energy barriers between them higher, which is a challenge for EBM training especially if we want to recover the relative weights of these classes (as we do).
>
> Finally, let us mention that in the revised version we will include a more quantitative comparison between our proposed method and CD (PCD) for the MNIST experiment (including the test using the FID as metric, see Fig. 2  in the figure file the ***Common Reply***). In addition, we conducted additional tests on other synthetic examples involving high-dimensional discrete datasets (see the description given in the ***Common Reply*** and Fig. 3  in the figure file).
>
> **Role of continuous-time analysis.** We believe that the continuous-time analysis will be transparent to the reader familiar with the statistical physics literature and will help them build the intuition for the discrete-time results. For example, the proof of Proposition 1 immediately shows why the resampling step is theoretically consistent (see also our reply to Reviewer 1pft).
>
> **Statements requiring references and justifications.**
> * *Among these methods, EBMs are perhaps the simplest and most interpretable.* We believe that this stems from the fact that these models benefit from a century of intuitions from computational physics to guide the design and interpretation of the energy $U$ and help the sampling of its associated Gibbs distribution. This is stated in the introduction already.
> * *Methods in this class are provably unable to learn the relative weights of modes when they are separated by low-density regions.* This is a known issue, discussed e.g. in Sec. 4.4 of the review paper by Song & Kingma, *How to Train Your Energy-Based Models*, arXiv:2101.03288.  Mathematically, it can be shown that the metric induced by the score is weaker than the one induced by KL since the former is related to the Hessian of the latter.
> * *Unlike the Fisher divergence, the cross-entropy is sensitive to the relative weights in multimodal distributions, but it unfortunately leads to training algorithms that are less justified theoretically and more delicate to use in practice.* The main issue is the computation of the KL and its gradient, which is the problem that we address in our paper.
> * *Unfortunately, the approximations made in these algorithms are uncontrolled and they are known to induce biases similar to those observed with the score-based methods.*  Because these methods (unlike ours) do not perform stochastic gradient descent (SGD) on the KL divergence, it is difficult to analyze their convergence properties and the nature of their fixed point.
> * *"Score-matching techniques and variants originate from [9, 21, 22]; their shortcoming in the context of EBM training is investigated in [23] and their blindness to the presence of multiple, imbalanced modes in the target density has been known for long and we refer to [24, 25]) for discussions." In this context, it is necessary to mention [4] (and numerous papers based on it), which is a score-based technique that led to a number of state-of-the-art methods in generative modeling, at the same time, sharing many common properties with EBMs (e.g. density evaluation).* We thank the reviewer for pointing out these references to us: we will cite them in the new version of our paper. As far as we can tell, these papers propose variants of EBM training that combines it with other techniques, but they do not guarantee that these methods perform training by effectively minimizing the KL divergence. We believe that our method could be used in concert with some of these approaches to remedy this deficiency.
>
> **Limitations:**
> * *Batch size:* In our new numerical experiments we show that the batch size can be reduced, and a gain in efficiency can also be achieved by using mini-batching (see also the reply to Reviewer d8UC).  In particular, we performed experiments with GMM using Algorithm 4 in the paper with mini-batches of various sizes over a total population of $10^4$, showing that the method performs well even with small mini-batches see Fig. 1 in the common reply. Similarly, we did experiments with MNIST using Algorithm 4 with a mini-batch size of $128$ samples over a sample size of $1024$. The performance is given in Fig. 2 of our common reply. The training time of different methods is provided in the caption of Fig. 2. These results show that our method is scalable with mini-batching, which accelerates training without much sacrifice of the performance. For more details on the values of hyperparameters that we used for MNIST experiments, please see our reply to Reviewer d8UC.
>
> * *Biased estimator:* Indeed our estimator is biased. This is a common issue with estimators used in the sequential Monte Carlo method. We did not observe any noticeable effect of this bias for the number of walkers we used.

---

> > ### Comment · Reviewer_ripa · 2023-08-12
> >
> > Thank you for your response. The current rebuttal does not address any of my concerns.

---

> > > ### Author Response · Authors · 2023-08-16
> > >
> > > We thank the reviewer for replying to us. We would appreciate if they could be more specific on how to address their concerns. Among generative models, EBM have many advantageous features that make them both versatile and interpretable, as nicely discussed e.g. in https://arxiv.org/abs/1903.08689. This reference along with the other mentioned by the reviewer discuss many clever tricks to improve  how to train of EBM and use the as generative models. Most of these tricks (e.g. replay buffer, adding Gaussian noise to data, data augmentation) are not yet justified theoretically, however, mainly because they aim to train EBM via SGD on the KL divergence but offer no guarantee to achieve this. In contrast, our approach offers these guarantees and it does so at not significant extra cost: as a result **our approach can potentially be used in concert with the improved methods discussed in the references quoted by the reviewer, by adding weights to the MCMC walkers as our paper explains**. We are now in the process of testing our approach on CIFAR10, as suggested by several reviewers, and we hope that these results will be available very soon, in which case we will include them in the paper. At the same time, **we believe that the theoretical results we derive, and the numerical  experiments we already have, already make  a case for our approach that is strong enough to justify publication in NeurIPS.**

---

> > > > ### Comment · Reviewer_ripa · 2023-08-19
> > > >
> > > > Dear authors, I totally agree that the proposed approach has clear theoretical benefits, and I, personally, sympathize with it more than the numerous ad-hoc solutions developed by the community previously. However, unfortunately, I cannot recommend the paper for acceptance without clear evidence of practical benefits or, at least, extensive empirical study (without benefits). The current developments could be presented at one of the numerous NeurIPS workshops, and I'm sure that the extended version could be later published at one of the major conferences.

---

> > > > ### Author Response · Authors · 2023-08-20
> > > > **New experiment on the CIFAR-10 image dataset**
> > > >
> > > > **CIFAR-10 Experiments**
> > > > Following the Reviewers suggestion, we performed an empirical evaluation of our method on the full CIFAR-10 (32 $\times$ 32) image dataset. We used the setup of Ref. [1], with the same neural architecture with $n_f = 64$ features, and compared the results obtained with our approach with mini-batching (Algorithm 4) to those obtained with PCD and PCD with data augumentation of Ref. [2] (which consists of a combination of color distortion, horizontal flip, rescaling, and Gaussian blur augumentations, to help the mixing of the MCMC sampling and stabilize training).
> > > >
> > > > The code used to performed these new experiments is available (along with some generated images) in the anonymized github referenced in our paper. The hyperparameters are the same in all cases: we take $N = 4096$ Langevin walkers with a mini-batch size $N' = 256$. We use the Adam optimizer with learning rate $\eta = 10^{-4}$ and inject a Gaussian noise of standard deviation $\sigma = 3\times 10^{-2}$ to the dataset while performing gradient clipping in Langevin sampling for better performance. All the experiments were performed on a single A100 GPU. Training for 600 epochs took about 22 hours  with the PCD algorithm (w/ and w/o data augumentation) and about 23 hours with our method.
> > > >
> > > >
> > > > The results in terms of FID and IS are:
> > > > | Method     | FID   | IS
> > > > | -------    |------ |------
> > > > |PCD |45.63 | 5.62
> > > > |PCD with data augmentation |42.27| 6.34
> > > > |Our method (Algorithm 4)| 38.49 | 6.76
> > > >
> > > > We will add these new results to the revised version of our paper. They indicate that our method can achieve slightly better performance than the PCD algorithms w/ and w/o data augumentation at a similar computational cost. Furthermore, these results on CIFAR-10 suggest that our method scales well to complicated training tasks on more realistic data sets.
> > > >
> > > > ***References:***
> > > >
> > > > [1] Nijkamp, E., Hill, M., Zhu, S. C., & Wu, Y. N. (2019). Learning non-convergent non-persistent short-run MCMC toward energy-based model. Advances in Neural Information Processing Systems, 32.
> > > >
> > > > [2] Du, Y., Li, S., Tenenbaum, J., & Mordatch, I. (2020). Improved contrastive divergence training of energy-based models. arXiv preprint arXiv:2012.01316.

---

### Official Review · Reviewer_1pft · 2023-07-20

**Soundness:** 3 good
**Presentation:** 3 good
**Contribution:** 2 fair
**Rating:** 5
**Confidence:** 4

**Summary:**

This paper examines EBM learning using tools from the Jarzynski and Sequential Monte Carlo. The method is meant to address limitations of current EBM methods, which often rely on simulated Markov Chains to calculate approximate gradient of the log normalizer. Since the chains in practice do not converge to the model distribution, the theoretical validity of standard procedures is questionable and practical issues with correct multi-modal modeling can arise. The central contribution of this work is to introduce a new loss function for EBM learning. Using the Jarzynski equality, the gradient of the log normalizer is shown to be equivalent to the ratio of the expectation of weighted gradients of the EBM evaluated at ULA samples and the expectation of the weight terms. Unlike standard EBM learning which requires an unjustified assumption of MCMC convergence for validity, this proposed loss function is exact even for ULA samples which have not reached the model equilibrium distribution. To improve sample efficiency, weights are periodically resampled during practical implementation. Experiments on Gaussian mixture models and MNIST digits show the proposed method is better able to learn correct relative weightings of modes compared to standard EBM models.

**Strengths:**

* The theoretical development of the proposed loss function is the central strength of the paper. In particular, developing a loss function that is valid even for non-convergent ULA samples is a major contribution that has the potential to lead to more principled and effective EBM learning.
* The paper is clearly written and easy to understand.
* The proposed method does not significantly increase computation costs beyond standard EBM methods.
* A thorough appendix with full proofs of theoretical results and a significant amount of supporting information and additional experimental results is included.
* Experimental results give convincing evidence that the proposed method is able to model relative modes weightings more effectively than standard EBM learning.

**Weaknesses:**

* The primary weakness is the scope of the experimental results. The scale of datasets studied is smaller than typical EBM works, which usually examine at least CIFAR-10 as a baseline. In my experience, many methods which are effective for GMMs and MNIST might have poor performance on CIFAR-10. Since the goal of the proposed method is to improve multi-modal modeling, it is difficult to gauge the potential of the proposed method when experiments only examine datasets with very few modes. FID scores on CIFAR-10 would provide a clearer comparison of the generative abilities of the proposed method compared to related methods.
* The theoretical validity of weight resampling technique in Section 3.3 is not discussed. My guess is that this step is not theoretically consistent with the loss derivation, although I am not sure about this. The paper should discuss the validity of this step. If it is theoretically consistent with the loss function, then proof should be given. If not, this should be acknowledged as a limitation of the proposed method.

**Questions:**

* Can the performance of the proposed method on larger scale datasets (e.g. CIFAR-10) be investigated?
* Can the authors discuss the validity of the weight sampling step in Section 3.3?

**Limitations:**

Authors did not include explicit sections discussing limitations and broader impacts.

---

> ### Author Rebuttal · Authors · 2023-08-09
>
> We would like to thank the reviewer for their careful reading, positive view on the manuscript, and constructive suggestions. Below, we address the main points raised.
>
> **Scope of the experimental results.** As stated in the common reply, we did not have the computing resources necessary to perform experiments on large datasets such as CIFAR-10 or ImageNet, unfortunately. However, we plan to perform these experiments in collaboration with other groups with more computational resources in a follow-up paper.
>
> We also stress that our experiments on MNIST are nontrivial: indeed, by pruning the dataset to 3 digits, we make the overlap between the classes smaller and the energy barriers between them higher, which is a challenge for EBM training especially if we want to recover the relative weights of these classes (as we do).
>
> Finally, let us mention that in the revised version we will include a more quantitative comparison between our proposed method and CD (PCD) for MNIST experiments (including test using the FID as metric, see Fig. 2  in the figure file the ***Common Reply***). Though we do see that our method surpasses PCD in terms of the FID score, we want to mention that the FID score is not a good metric for MNIST images. In order to use the pre-trained Inception V3 network for the computation of the FID score, we copy the grayscale channel three times and resize the images from $28\times 28$ to $299\times 299$ pixels. Since there are limited results in the literature on benchmarking the training performance on MNIST images with the FID score, we do not report and compare results from other papers here. Since the Inception v3 network is trained and used for real images (e.g. CIFAR-10, ImageNet) and MNIST hand-written digits are not natural objects, we do not seek to get reasonably low FID scores for MNIST images. In addition to quantitative results on the MNIST dataset, we conducted additional tests on other synthetic examples involving high-dimensional discrete datasets (see the description given in the ***Common Reply*** and Fig. 3  in the figure file).
>
> **Validity of the resampling step.** *The resampling step is theoretically consistent with the computation of the gradient of the KL divergence used as the loss.* This can be seen from Eq. (10): the last term in this equation can be interpreted as a birth/death term whose effect is statistically equivalent (in the infinite number of walkers limit) to keeping the weights and can be implemented in practice via resampling. This duality between tracking the weights of the walkers, or resampling them according to these weights, is valid both in the discrete- and continuous-time setups, and is at the core of sequential Monte-Carlo methods where resampling is used to maintain a good effective sample size (ESS). These arguments are standard, but for the reader's convenience, we will include proof in the final version of our paper.
>
> **Limitations and broader impacts.** We will include a more thorough discussion of the main limitation of our method, namely that the number of walkers must be large enough to ensure statistical accuracy -- the need of using a large number of walkers is common to all EBM training methods, and the impact on efficiency can be mitigated by using mini-batches (as discussed in the answer to Reviewers ripa and d8UC). We will also discuss the broader impact of our approach more thoroughly, in particular emphasizing the widespread usage of EBM and the need to use methods that ensure fairness in this context (as ours does).

---

> > ### Comment · Reviewer_1pft · 2023-08-17
> > **Thanks for the author responses. I will keep my score.**
> >
> > Thanks to the authors for their responses to myself and other reviewers. I greatly appreciate the efforts of this work to develop an EBM learning framework that is less reliant on assumptions which are demonstrably not true in practice. This work has ideas which could be a step towards learning proper unnormalized densities. As the authors acknowledge, this method relies on approximations as well, namely the use of a finite number of walkers to approximate certain expectations, and the accuracy of these approximations in the finite sample case remain opaque. While one can speculate about the relative accuracy of the approximations for current EBM methods and the proposed method for high-dimensional data, rigorous comparisons cannot be made. Empirical results are the usual method comparison in this case, but this paper does not perform experiments at a scale that is sufficient for a fair comparison. I agree with Reviewer nHa5 that there are public resources for EBM learning which are very accessible with relatively small compute cost (especially given free/low-cost cloud GPU platforms). I do not view it as necessary to achieve state-of-the-art results on CIFAR-10 with the current method, but at least viable FID scores are needed to validate the claims in this paper.
> >
> > Overall, I choose to keep my score.

---

> > > ### Author Response · Authors · 2023-08-20
> > > **New experiment on the CIFAR-10 image dataset**
> > >
> > > **CIFAR-10 Experiments**
> > > Following the Reviewers suggestion, we performed an empirical evaluation of our method on the full CIFAR-10 (32 $\times$ 32) image dataset. We used the setup of Ref. [1], with the same neural architecture with $n_f = 64$ features, and compared the results obtained with our approach with mini-batching (Algorithm 4) to those obtained with PCD and PCD with data augumentation of Ref. [2] (which consists of a combination of color distortion, horizontal flip, rescaling, and Gaussian blur augumentations, to help the mixing of the MCMC sampling and stabilize training).
> > >
> > > The code used to performed these new experiments is available (along with some generated images) in the anonymized github referenced in our paper. The hyperparameters are the same in all cases: we take $N = 4096$ Langevin walkers with a mini-batch size $N' = 256$. We use the Adam optimizer with learning rate $\eta = 10^{-4}$ and inject a Gaussian noise of standard deviation $\sigma = 3\times 10^{-2}$ to the dataset while performing gradient clipping in Langevin sampling for better performance. All the experiments were performed on a single A100 GPU. Training for 600 epochs took about 22 hours  with the PCD algorithm (w/ and w/o data augumentation) and about 23 hours with our method.
> > >
> > >
> > > The results in terms of FID and IS are:
> > > | Method     | FID   | IS
> > > | -------    |------ |------
> > > |PCD |45.63 | 5.62
> > > |PCD with data augmentation |42.27| 6.34
> > > |Our method (Algorithm 4)| 38.49 | 6.76
> > >
> > > We will add these new results to the revised version of our paper. They indicate that our method can achieve slightly better performance than the PCD algorithms w/ and w/o data augumentation at a similar computational cost. Furthermore, these results on CIFAR-10 suggest that our method scales well to complicated training tasks on more realistic data sets.
> > >
> > > ***References:***
> > >
> > > [1] Nijkamp, E., Hill, M., Zhu, S. C., & Wu, Y. N. (2019). Learning non-convergent non-persistent short-run MCMC toward energy-based model. Advances in Neural Information Processing Systems, 32.
> > >
> > > [2] Du, Y., Li, S., Tenenbaum, J., & Mordatch, I. (2020). Improved contrastive divergence training of energy-based models. arXiv preprint arXiv:2012.01316.

---

### Author Rebuttal · Authors · 2023-08-09

We would like to thank all of the reviewers for their careful reading, constructive suggestions, and overall interest in our work. In this reply, we answer to the main questions common to all reviewers; below, we address their concerns individually and give more details on each point.

***Regarding the theory part:*** Our approach permits the calculation of the KL divergence and its gradient by adding scalar weights to the walkers used to do the sampling. This calculation is exact in the limit of infinitely many walkers and can be simulated using standard techniques from sequential Monte-Carlo. We stress that our method is no more costly than the CD or PCD algorithms. In particular (all these points are elaborated upon in our individual replies to the reviewers):
1. Our resampling step can be justified theoretically and is statistically equivalent to evolving the weights (assuming that the number of walkers is infinite). Since these weights are scalar, their storage and evolution involve no noticeable extra cost.
2. When the number of walkers is finite, the resampling step helps to improve the sample efficiency of our approach since this step allows us to control the effective sample size (which, with methods like CD or PCD, is harder to measure and control since its calculation involves the walkers weights that our method introduces).
3. The resampling step only involves selecting walkers using their weights. Since these weights are scalar quantities, the cost of this step is independent of the dimensionality of the problem.
4. Our method can easily be used with mini-batches, in which only a subset of the walkers are evolving at any given time, and the rest are kept in a replay buffer, see Algorithm 4 in the paper. In fact, as we explain in our reply to Reviewer d8UC, using mini-batches with our approach is better justified than doing so in the PCD algorithm. To emphasize this point better, we will report experiments with mini-batches both in the GMM example (see Fig. 1 in the figure file) as well as MNIST (see Fig. 2).


***Regarding the numerical experiments:*** We agree with the reviewers that our method should be tested on larger and more realistic datasets such as CIFAR-10 or ImageNet but unfortunately we do not have the computing resources necessary to perform these experiments. We plan to perform these simulations in collaboration with other groups with more computational resources in a follow-up paper. We also believe that the strong theoretical underpinning of our approach is sufficient to justify its interest for the NeurIPS community.

We also stress that our experiments on MNIST are nontrivial: indeed, by pruning the dataset to 3 digits, we make the overlap between the data classes smaller, which is a challenge for EBM training especially if we want to recover the relative weights of these classes (as we do). To emphasize this point,  in the revised version we will include a more quantitative comparison between our proposed method and CD (PCD) for our MNIST experiment (see Fig. 2 in the figure file).

Finally, we conducted additional tests on other synthetic examples involving high-dimensional discrete datasets (see the description given below and Fig. 3 in the figure file). These new tests show again that our method performs better than the CD and PCD algorithms at a similar numerical cost. (They also show that our approach can be straightforwardly generalized to EBM defined on discrete datasets, for example graph-structured datasets, on which we plan to test our method on subsequent works)

**Experimental setup of the synthetic experiment involving discrete datasets:**
We used the experimental setup of Refs [1,2]. We first generate synthetic 2D data $x \in \mathbb{R}^2$ from classical datasets (eg, Moons, 8Gaussians, etc). Then the floating-point number representation of $x$ (with precision up to $\texttt{1e-4}$ ) is turned into a $32$-bit code constructed using concatenated Gray codes (https://en.wikipedia.org/wiki/Gray\_code), which results in a new distribution on the discrete space $\\{0,1\\}^{32}$. The transformation from $\mathbb{R}^2$ to $\\{0,1\\}^{32}$ is challenging since Gray codes are not linear.
The discrete space has size $2^{32}$  (extremely large). We also introduce variants of the datasets where different modes are assigned different weights (Figure 4 in the figure file). Our codebase directly iterates on the ones found in Refs. [1,2].

This experimental setup allows nice visualizations (in the 2D space): visual analysis of the trained models is performed using the heatmaps of the learned energy functions (see Fig. 4 in the figure file). As a metric, we use Maximum Mean Discrepancy with a linear kernel over $\\{0,1\\}^{32}$ (corresponding to the Hamming distance) to compare ${4000}$ samples from the data distribution and $4000$ samples from our model.

***Hyperparameters:*** We parametrize our EBMs with a 4-layer MLP, ELU activations, and hidden layer size $256$. We did not optimize hyperparameters and directly used the ones from Refs. [1,2]: we use the Adam optimizer with a learning rate of $\texttt{3e-4}$, weight decay of $\texttt{1e-4}$, gradient clipping at 1, batch size of $128$ and $100$ batches per training epoch, with $1000$ training epochs.

The results in Figure 3 (in the figure file) show the competitiveness of our method. Figure 4 additionally shows how our method is able to retrieve the correct mode masses on imbalanced, multimodal datasets.

**References**
[1] Hanjun Dai, Rishabh Singh, Bo Dai, Charles Sutton, and Dale Schuurmans. Learning discrete energy-based models via auxiliary-variable local exploration. Advances in Neural Information Processing Systems, 33:10443–10455, 2020.

[2] Meng Liu, Haoran Liu, and Shuiwang Ji. Gradient-guided importance sampling for learning binary energy-based models. arXiv preprint arXiv:2210.05782, 2022.

---

> ### Author Response · Authors · 2023-08-20
> **New experiment on the CIFAR-10 image dataset**
>
> **CIFAR-10 Experiments**
> Following the Reviewers suggestion, we performed an empirical evaluation of our method on the full CIFAR-10 (32 $\times$ 32) image dataset. We used the setup of Ref. [1], with the same neural architecture with $n_f = 64$ features, and compared the results obtained with our approach with mini-batching (Algorithm 4) to those obtained with PCD and PCD with data augumentation of Ref. [2] (which consists of a combination of color distortion, horizontal flip, rescaling, and Gaussian blur augumentations, to help the mixing of the MCMC sampling and stabilize training).
>
> The code used to performed these new experiments is available (along with some generated images) in the anonymized github referenced in our paper. The hyperparameters are the same in all cases: we take $N = 4096$ Langevin walkers with a mini-batch size $N' = 256$. We use the Adam optimizer with learning rate $\eta = 10^{-4}$ and inject a Gaussian noise of standard deviation $\sigma = 3\times 10^{-2}$ to the dataset while performing gradient clipping in Langevin sampling for better performance. All the experiments were performed on a single A100 GPU. Training for 600 epochs took about 22 hours  with the PCD algorithm (w/ and w/o data augumentation) and about 23 hours with our method.
>
>
> The results in terms of FID and IS are:
> | Method     | FID   | IS
> | -------    |------ |------
> |PCD |45.63 | 5.62
> |PCD with data augmentation |42.27| 6.34
> |Our method (Algorithm 4)| 38.49 | 6.76
>
> We will add these new results to the revised version of our paper. They indicate that our method can achieve slightly better performance than the PCD algorithms w/ and w/o data augumentation at a similar computational cost. Furthermore, these results on CIFAR-10 suggest that our method scales well to complicated training tasks on more realistic data sets.
>
> ***References:***
>
> [1] Nijkamp, E., Hill, M., Zhu, S. C., & Wu, Y. N. (2019). Learning non-convergent non-persistent short-run MCMC toward energy-based model. Advances in Neural Information Processing Systems, 32.
>
> [2] Du, Y., Li, S., Tenenbaum, J., & Mordatch, I. (2020). Improved contrastive divergence training of energy-based models. arXiv preprint arXiv:2012.01316.

---

> ### Author Response · Authors · 2023-08-21
>
> As the discussion period comes to an end, we wish to thank all the reviewers for their constructive feedback on our paper. We made every effort to address their concerns, in particular about the scalability of our method and the scope of our numerical experiments. We hope that our new results on the CIFAR-10 image dataset will alleviate their worries on this front and lead them to reassess their scores.

---

### Comment · Area_Chair_TauV · 2023-08-13

Dear reviewers and authors,

Thank you very much for your work on this submission and its evaluation. Now that the authors have responded to the reviews, I strongly encourage the reviewers to acknowledge the review, to look at other reviews and rebuttals for this submission, and to adjust their scores if needed. Thanks to those that have already done so.

Authors have the possibility to reply if further questions are needed, until the 16th.

Thank you very much to all,
Area Chair

---

### Decision · Program_Chairs · 2023-09-21

**Decision:**

Accept (poster)

**Comment:**

The authors proposed a modification of the standard procedure to train an energy-based model. Their method is based on Jarzynski equality and is theoretically justified.

Initially, to demonstrate efficiency of the approach the authors consider GMMs and MNIST dataset, which is not sufficient from my point of view. However, during the rebuttal, the authors demonstrated efficiency also using CIFAR10 dataset. I think this is enough to justify that the method actually works.

Overall, taking into account other positive comments from reviewers I vote to accept the paper.
At the same time, it would be very important to take carefully all comments of the reviewers when finalising the paper.